# A rapid whisker-based decision underlying skilled locomotion in mice

**Richard A Warren\*, Qianyun Zhang, Judah R Hoffman, Edward Y Li, Y Kate Hong†, Randy M Bruno, Nathaniel B Sawtell\***

Department of Neuroscience, Mortimer Zuckerman Mind Brain Behavior Institute, Columbia University, New York, United States

**Abstract** Skilled motor behavior requires rapidly integrating external sensory input with information about internal state to decide which movements to make next. Using machine learning approaches for high-resolution kinematic analysis, we uncover the logic of a rapid decision underlying sensory-guided locomotion in mice. After detecting obstacles with their whiskers mice select distinct kinematic strategies depending on a whisker-derived estimate of obstacle location together with the position and velocity of their body. Although mice rely on whiskers for obstacle avoidance, lesions of primary whisker sensory cortex had minimal impact. While motor cortex manipulations affected the execution of the chosen strategy, the decision-making process remained largely intact. These results highlight the potential of machine learning for reductionist analysis of naturalistic behaviors and provide a case in which subcortical brain structures appear sufficient for mediating a relatively sophisticated sensorimotor decision.

**\*For correspondence:**
richardwarren2163@gmail.com
(RAW);
ns2635@columbia.edu (NBS)

**Present address:** †Carnegie
Mellon University, Department of
Biological Sciences and Carnegie
Mellon Neuroscience Institute,
Pittsburgh, United States

**Competing interests:** The
authors declare that no
competing interests exist.

**Reviewing editor:** Jesse H
Goldberg, Cornell University,
United States

## Introduction

Perception, decision-making, and motor control are often serialized in the lab, with animals collecting sensory information over time before responding with discrete actions from static resting positions. Despite the utility of this approach (*Gold and Shadlen, 2007*; *Shenoy, 2011*; *Carandini and Churchland, 2013*; *Svoboda and Li, 2018*), it does not capture many real-world behaviors in which perception, decision-making, and action occur rapidly and in parallel (*Cisek and Kalaska, 2010*). For example, in deciding how to avoid an obstacle in its path (e.g. by breaking, turning, or stepping over it), an animal must consider the size and position of the obstacle, the position and velocity of the limbs, and respond such that the obstacle is avoided while maintaining balance. The speed with which such decisions are made may preclude strategies based on the gradual accumulation of evidence that are often associated with cerebral cortex (*Shadlen and Kiani, 2013*). In terrestrial locomotion, sensorimotor decisions are further complicated by the high dimensionality of the musculoskeletal system. Whereas cognitive and perceptual decisions often have categorical or binary outcomes (e.g. which action to take or whether a stimulus is present), decisions about movement must coordinate multiple limbs in a manner that respects ongoing changes to the state of the body (*Drew et al., 2004*; *Marigold and Drew, 2017*).

Sensorimotor decisions can mean life or death for animals catching prey or escaping predators. However, with some notable exceptions (see e.g. *Branco and Redgrave, 2020*), the behavioral strategies and underlying neural mechanisms are not well characterized. One bottleneck has been the technical difficulty of performing thorough behavioral analysis of sensorimotor decisions in the context of whole-body behaviors such as locomotion. Until recently, detailed kinematic descriptions of even 'simple' behaviors such as locomotion required elaborate tracking systems involving physical markers attached to animals' joints (*Belanger et al., 1996*; *Austin et al., 1999*; *Kaya et al., 2003*; *Aoki et al., 2012*), which is particularly challenging for small animals such as mice (*Akay et al., 2014*; *Setogawa et al., 2015*). These hurdles have been largely overcome by modern machine

learning tools such as convolutional neural networks (CNNs) (*LeCun et al., 2015*), which accurately track body pose without markers (*Mathis et al., 2018*; *Graving et al., 2019*; *Pereira et al., 2019*). Combining modern machine vision algorithms with high-speed, multi-view imaging allows automatic three-dimensional analyses with high spatial and temporal resolution (*Machado et al., 2015*; *Nath et al., 2019*), thus facilitating comprehensive exploration of complex behaviors and their computational underpinnings (*Anderson and Perona, 2014*; *Dell et al., 2014*; *Krakauer et al., 2017*; *Datta et al., 2019*).

We leveraged these recent advances to perform a detailed kinematic analysis of sensory-guided locomotion in head-fixed mice. We show that mice rely on whisker (rather than visual) input to shape limb trajectories while stepping over obstacles at high speeds. This behavior entails a rapid sensori-motor decision in which whisker information and locomotor state are integrated to drive distinct kinematic strategies. These strategies remain largely intact after perturbations of either primary somatosensory or primary motor cortex, suggesting that the decision is made subcortically.

## Results

### Obstacle clearance during high-speed locomotion in head-fixed mice

We developed a head-fixed sensory-guided locomotion task compatible with high-throughput, three-dimensional behavioral tracking (*Video 1*). Using a custom running wheel (*Warren and Hoffman, 2018*) with a transparent floor and a mirror mounted inside at 45°, we reconstructed the body pose in three dimensions with a high-speed camera (*Figure 1A–B*). DeepLabCut software (*Mathis and Warren, 2018*; *Mathis et al., 2018*) was used to automatically track the paws and tail with high accuracy (*Figure 1B*; Materials and methods). We also developed custom neural network tools to determine when the paws and whiskers contacted the obstacle (the latter was determined using an additional high-speed camera focused on the whiskers) (*Figure 1B–C*, *Figure 1—figure supplement 1F–K*; Materials and methods).

Mice ran in a dark enclosure, receiving a water reward every 5.4 m. Their running intermittently activated an illuminated cylindrical obstacle ~30 cm away that moved towards them at a speed matching that of the wheel (*Figure 1A*; *Figure 1—figure supplement 1A–B*). Hence, this closed-loop setup simulates stepping over a stationary object. The obstacle was visible for several hundred milliseconds before it reached the mouse (0.67 ± 0.39 s depending on the speed of locomotion), such that mice had the opportunity to make visually driven preparatory gait modifications like those described in other species (*Drew et al., 1996*; *Higuchi, 2013*). To reduce the predictability of the obstacle's position, on each trial we randomized the obstacle height (4–10 mm), the wheel distance necessary to engage the obstacle (1.8 ± 0.1 m), and the distance of the obstacle to the mouse when it started moving (0.31 ± 0.015 m).

Some mice stepped over the obstacle from the very first trial, whereas others tended to grasp it. We therefore adopted a training regimen to discourage grasping in which contact with the obstacle triggered a break of the wheel and a loud auditory stimulus (Materials and methods). After training (~2 weeks), mice successfully cleared the obstacle on a large fraction of trials irrespective of obstacle height (*Figure 1D*; success defined as <= 20 ms of paw contact with the obstacle in a trial).

Notably, mice maintained high running speeds even as they stepped over the obstacle (*Figure 1E*, *Figure 1—figure supplement 1A*; *Video 2*). In past research in humans (*van Hedel et al., 2002*), cats (*Drew et al., 1996*), rats (*Sato et al., 2012*; *Setogawa et al., 2015*), and mice (*Asante et al., 2010*), obstacle avoidance was studied at substantially slower speeds associated with walking gaits. Mice in our paradigm ran ~5 times faster than the mice in *Asante et al., 2010* and ~30% faster than the

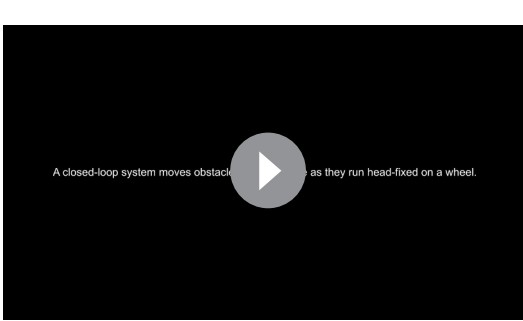

**Video 1.** Demonstration of the head-fixed obstacle avoidance apparatus, 3D behavioral tracking, and kinematic analysis.
https://elifesciences.org/articles/63596#video1

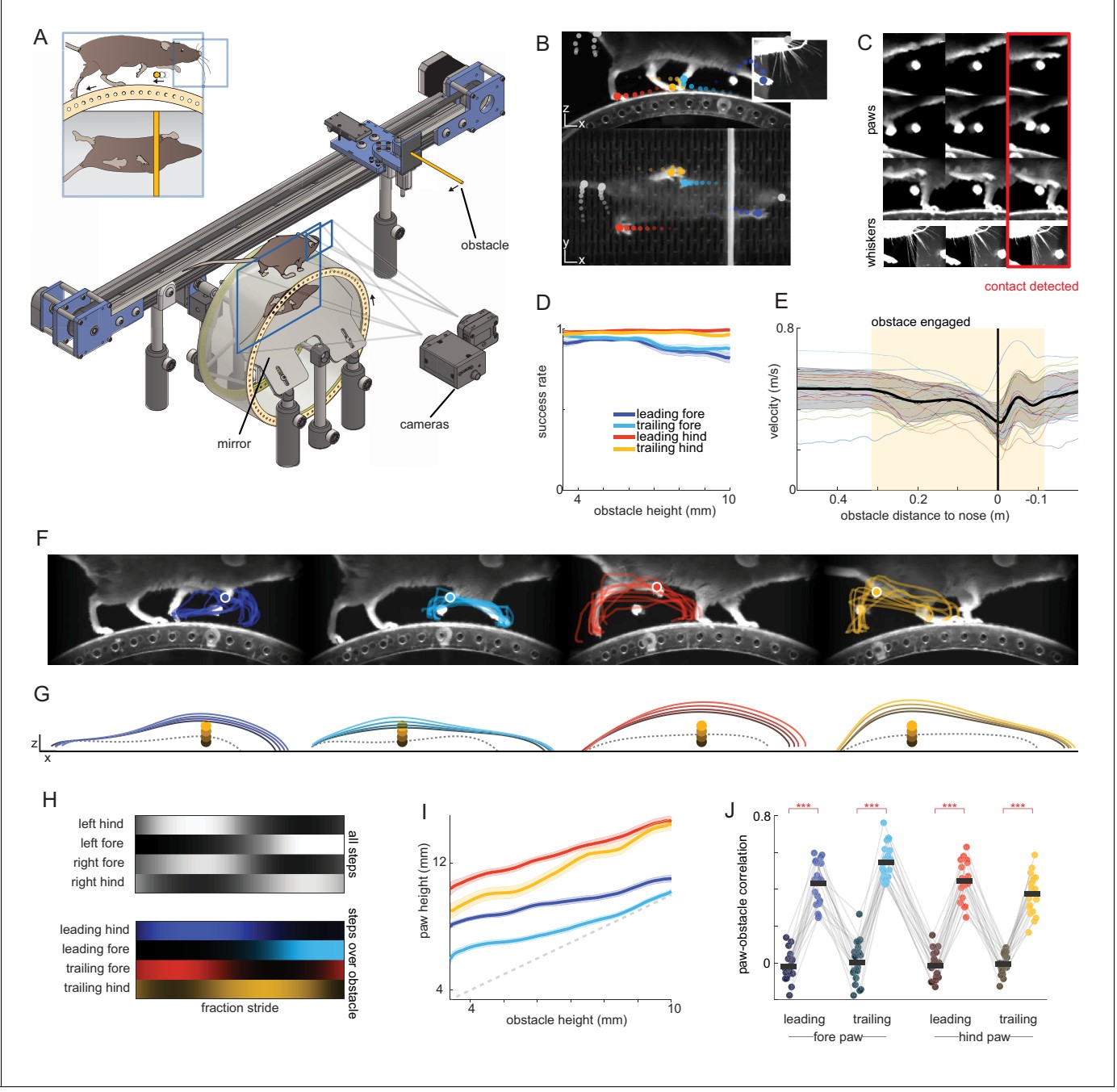

**Figure 1.** Head-fixed obstacle avoidance in mice. (**A**) Schematic of the obstacle avoidance setup. A head-fixed mouse runs on top of a wheel with a mirror mounted inside, allowing a single camera to capture two orthogonal views. A second camera focuses on the whiskers. An obstacle is moved toward the mouse along a motorized, belt-driven linear track at a speed matched to that of the wheel. (**B**) DeepLabCut is used to track the positions of the paws, tail, and nose in both views. Tracking from the two views is combined to reconstruct the three-dimensional pose at 250 Hz. (**C**) Custom convolutional neural network algorithms were developed to determine when the whiskers (bottom row) and paws (top rows) contact the obstacle. (**D-J**) Behavioral characterization of head-fixed obstacle clearance for n = 20 mice. (**D**) Obstacle clearance success rate as a function of obstacle height for each paw (mean with S.E.M. shaded). All paws cleared the obstacle at high rates even for high obstacles. (**E**) Average running velocity as a function of position relative to the obstacle (standard deviation is shaded; positive numbers mean the obstacle is in front of the mouse). Thin lines are averages for individual mice. Vertical black line shows the position at which the obstacle is beneath the nose of the mouse. (**F**) Example trial showing (from left to right) the leading forelimb (LF), trailing forelimb (TF), leading hindlimb (LH), and trailing hindlimb (TH) clearing the obstacle. Traces show kinematics from trials selected randomly from a single session. (**G**) Average kinematics across mice for LF, TF, LH, and TH binned by obstacle height (colored traces). Dashed gray traces are the average of the two steps preceding whisker contact with the obstacle. The obstacle is 3.175 mm in diameter. (**H**) Hildebrand plots (*Hildebrand, 1989*) , averaged across mice, reveal trot gaits during normal locomotion and obstacle clearance. Color intensity

*Figure 1 continued on next page*

*Figure 1 continued*

represents likelihood of stance. The top panel is calculated across all steps, and the bottom panel from steps over the obstacle. (I) Paw height vs. obstacle height averaged across mice (S.E.M. is shaded). Height is measured when the paw is 8 mm in front of the obstacle. The dashed gray line is the unity line. (J) Average correlation between the obstacle height and the height of all paws, measured when the paw is 8 mm in front of the obstacle. Circles represent individual mice. For each paw, the correlation is computed for the step over the obstacle (colored circles), and the preceding control steps (dark circles). See also *Figure 1—figure supplement 1*.

The online version of this article includes the following figure supplement(s) for figure 1:

**Figure supplement 1.** Head-fixed obstacle avoidance in mice.

cats in *Drew et al., 1996*. Clearing obstacles at high speeds may require behavioral strategies and neural mechanisms that are distinct from those studied previously.

We characterized the three-dimensional kinematics of all four paws as they cleared the obstacle. To allow direct comparison with past studies in freely moving rodents, we analyzed kinematics in 'un-head-fixed' coordinates by subtracting the displacement of the wheel from the positional measurements (*Video 1*). Kinematics thus represent locomotion as if mice were moving forward in space. Kinematic analysis revealed that mice ran mostly in a trot pattern, wherein diagonal pairs of limbs move together but are antiphase with the opposite pair (*Figure 1H*; *Bellardita and Kiehn, 2015*; *Machado et al., 2015*). Consistent with previous studies of obstacle avoidance in freely moving rodents (*Aoki et al., 2012*; *Sato et al., 2012*; *Setogawa et al., 2015*), this pattern was usually maintained during obstacle clearance, such that paws cleared the obstacle sequentially: leading forelimb (LF), trailing forelimb (TF), leading hindlimb (LH), and trailing hindlimb (TH) (*Figure 1F–H*).

Mice made large adjustments to the trajectories of all paws as they cleared the obstacle, lifting them 2–3 times higher (*Figure 1—figure supplement 1D*; $p<10^{-10}$ for all paws) and extending them further (*Figure 1—figure supplement 1E*; LF, $p<0.05$; TF, $p<0.001$; LH, $p<0.01$; TH, $p<0.01$) than control steps (*Figure 1G*; control steps [dashed gray traces] are those that occurred prior to whisker contact with the obstacle). Furthermore, we found a clear relationship between obstacle height and paw height, with mice stepping higher to clear higher obstacles (*Figure 1G,I,J*; *Asante et al., 2010*; *Aoki et al., 2012*; *Sato et al., 2012*). Collectively, these results indicate that head-fixed mice perform a rapid sensorimotor transformation in which information about the location and height of an obstacle is transformed into graded adjustments of the kinematics of all four limbs to avoid collision.

## Obstacle clearance is whisker dependent

Although research on sensory-guided locomotion in humans and cats has focused largely on vision (*Drew et al., 1996*; *McVea and Pearson, 2009*), rodents may rely on additional sensory modalities such as whisker-mediated somatosensation (*Kleinfeld et al., 2006*; *Grant et al., 2012*). Hence, we next explored the contributions of both whiskers and vision to head-fixed obstacle clearance (experiment summarized in *Video 3*).

Mice were trained and tested with the obstacle illuminated on half of the trials (randomly interleaved) and the other half occurring in complete darkness (*Figure 2A*). When the obstacle could be sensed with both the eyes and whiskers (as in *Figure 1*), mice successfully cleared it at high rates (*Figure 2B*, green), adjusted the height of their paws corresponding to the height of the obstacle (*Figure 2C–D*, *Figure 2—figure supplement 1B*), and ran at high speeds (*Figure 2E*). However, they slowed down as they approached the obstacle (*Figure 2E*). Remarkably, without visual input mice maintained this level of performance without slowing down (*Figure 2E*). In the absence of visual input mice cleared the obstacle at similar rates (*Figure 2B*; p=0.19), matched the height of the paws to the

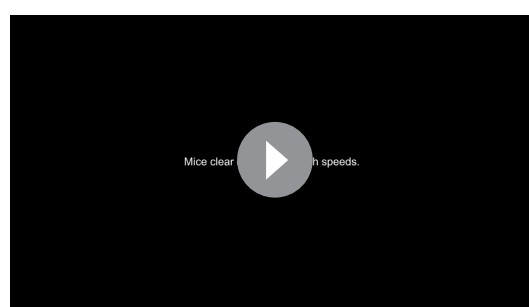

**Video 2.** Real-time and slowed-down videos of obstacle avoidance.
https://elifesciences.org/articles/63596#video2

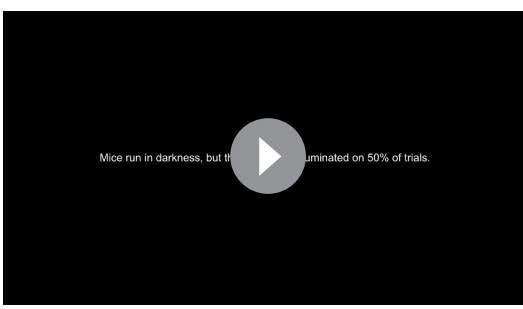

**Video 3.** Comparison of obstacle avoidance when mice have access to whiskers and vision, whiskers only, vision only, and neither whiskers nor vision.
https://elifesciences.org/articles/63596#video3

height of the obstacle to a similar extent (*Figure 2C–D*, *Figure 2—figure supplement 1B*; p=0.57 for the leading forepaw), and maintained high speeds even as they cleared the obstacle (*Figure 2E*).

We next tested whether mice require whiskers to perform the task. All whiskers were trimmed on both sides of the face after mice reached steady-state performance. Although the trident whisker could not reach the obstacle, we trimmed it as well. When the obstacle was visible but mice had no whiskers, success rates dropped (*Figure 2B*; p<0.01) despite no significant change in the average height of steps over the obstacle (*Figure 2C*, *Figure 2—figure supplement 1A*). Rather than affecting overall step height, whisker trimming abolished the correlation between the height of the leading forepaw and that of the obstacle (*Figure 2C–D*, *Figure 2—figure supplement 1B*; p<0.001), suggesting that whiskers – rather than vision – are necessary for estimating obstacle height. In a separate set of experiments, performance was assessed as whiskers were gradually trimmed. The ability to adjust the paw height to the height of the obstacle required more than one whisker (*Figure 2—figure supplement 1D*), and the accuracy of paw landing positions deteriorated as more whiskers were trimmed (*Figure 2—figure supplement 1E*). These results suggest mice combine information from multiple whiskers to determine both the height and horizontal position of the obstacle.

Whereas whiskers provide high-fidelity information about nearby objects, vision may be used to detect objects at a distance and drive preparatory changes. In humans, for example, vision is thought to guide positioning of the trailing foot at an appropriate distance relative obstacles (*Chou and Draganich, 1998*; *Patla and Greig, 2006*). Interestingly, mice with whiskers but no vision positioned their trailing forepaw more accurately (with less variability) compared to mice with vision only (*Figure 2F*, *Figure 2—figure supplement 1C*; p<0.05), suggesting that although whisker information only becomes available at the last moment, mice can quickly respond with accurate modifications.

Finally, with whiskers trimmed and lights off, mice ran at least as fast relative to the vision and whiskers condition (*Figure 2E*; p=0.11). However, mice were no longer able to successfully clear the obstacles (*Figure 2B*; p<0.01) or match the height of their paws to the height of the obstacle (*Figure 2C–D*, *Figure 2—figure supplement 1B*), ruling out roles for other sensory modalities. Overall, these results demonstrate that whisker somatosensation is sufficient to drive rapid behavioral modifications during obstacle clearance.

## A rapid sensorimotor decision underlies obstacle clearance

Relying on whisker input to clear obstacles at high speeds seemingly poses a challenge. At the moment of whisker contact, a paw will intercept the obstacle within ~63 ms if no modifications are made, and the closest paw is only ~32 mm away from the obstacle (pooled across trials with and without vision) (*Figure 3—figure supplement 1A*; Materials and methods). Moreover, the state of the body is highly variable across trials at whisker contact (*Figure 3—figure supplement 1B*), implying that mice must rapidly integrate whisker sensory input with information about the state of the body to execute appropriate responses.

Kinematic analysis of many trials revealed distinct strategies mice use to clear the obstacle (*Figure 3A–B*; *Video 4*). We focus our analysis on the forepaw in swing at whisker contact, as this paw is in the most immediate danger of colliding with the obstacle absent kinematic modifications. On some trials, this paw continues along its expected trajectory, whereas on other trials the step is shortened or lengthened relative to control steps. On shortened trials, the paw is placed in front of the obstacle such that the opposite paw can step over first, and on lengthened trials the paw usually (on 74% of trials) clears the obstacle in one large step. These step modifications were initiated rapidly. Within less than 30 ms of whisker contact the kinematic trajectory of the forepaw changed

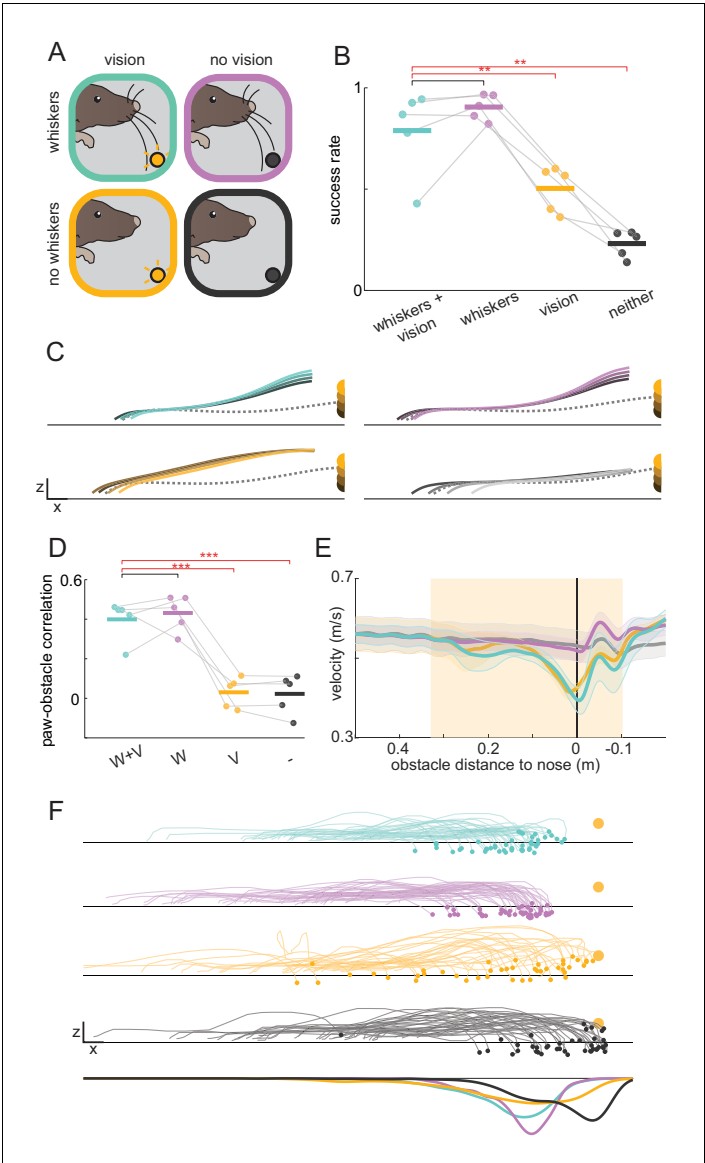

**Figure 2.** Mice rely on whiskers to clear obstacles. (**A**) Schematic of experimental paradigm. Obstacle avoidance was tested with whiskers (top row) and with trimmed whiskers (bottom row) in separate sessions. Within each session, randomly interleaved trials occurred in complete darkness (no vision, right column) or with the obstacle illuminated internally (vision, left column). (**B**) Obstacle clearance success rates when mice had access to whiskers and vision (W+V), whiskers without vision (W), vision without whiskers (V), or neither vision nor whiskers (-) (n = 5 for this and subsequent panels). (**C**) Kinematic trajectories of the leading forepaw binned by obstacle height for each sensory condition. Each line is an average across individual mouse averages. The dashed gray line is the average of the two steps preceding whisker contact with the obstacle. Kinematics are truncated 8 mm in front of the obstacle to demonstrate that height shaping emerges before paws can contact the obstacle. Height shaping therefore does not result from paw contacts. (**D**) Mice adjust the height of their leading forepaw based on the height of the obstacle only when whisker sensory information is available. The correlation between paw and obstacle height is measured when the paw is 8 mm in front of the obstacle. (**E**) Average wheel velocity binned by the position of the mouse relative to the obstacle (shaded lines are standard error; positive numbers mean the obstacle is in front of the mouse). The shaded box shows when the obstacle is engaged, and the vertical black line is the position at which the obstacle is beneath the nose of the mouse. (**F**) The landing position of the trailing forepaw is less variable when whisker sensory information is available. The top four rows show the kinematics of the step preceding the step over the obstacle for the lagging forepaw. Each trace is a single trial selected randomly across mice. The bottom row shows the distribution of landing positions pooled across mice. Each

*Figure 2 continued on next page*

*Figure 2 continued*

mouse was tested for two sessions with and two sessions without whiskers. See also *Figure 2—figure supplement 1*.

The online version of this article includes the following figure supplement(s) for figure 2:

**Figure supplement 1.** Mice rely on whiskers to clear obstacles.

significantly (*Figure 3—figure supplement 1C*; Materials and methods). These strategies were no longer apparent with whiskers trimmed and lights off (*Figure 3G*, bottom). Both strategies emerged with or without vision (as long as whiskers were present; *Figure 3G*); we therefore pooled both trial types for these analyses.

We explored whether the strategy used on a given trial (lengthening vs. shortening the step of the paw in swing at whisker contact) is systematically related to the state of the body and/or the position of the obstacle. We identified features of the body state and obstacle position that are predictive of whether mice shortened or lengthened their step (omitting trials where no modifications were made [Materials and methods]) by sequentially adding them to logistic Generalized Linear Models (GLMs) based on their ability to improve the models' accuracy (Materials and methods). Consistent with a deterministic decision-making process, models predicted the behavioral strategy with 73.0% accuracy using eight predictors (*Figure 3C*). Artificial neural networks trained with the same predictors had comparable accuracy of 73.1%, suggesting that the decision underlying this behavior may obey a relatively simple logic. Mice were less successful on trials in which their decision violated the predictions of the model (*Figure 3—figure supplement 1E*; p<0.001), suggesting that correct decision-making facilitates obstacle clearance.

The top features selected by the model suggest that the decision is influenced by both the state of the body and the position of the obstacle (*Figure 3C,D*). The top two features were obstacle proximity (the horizontal position of the obstacle at whisker contact; blue) and the horizontal paw position (orange). Mice are more likely to lengthen their step when whisker contact occurs early in the swing phase (when the horizontal paw velocity [purple] is low and the horizontal paw position [orange] is further back) and when mice are running faster (yellow). Furthermore, mice are less likely to lengthen their step when the obstacle is high (magenta) and far away (blue).

To better understand why mice lengthen or shorten their step on a given trial, we plotted kinematics for the forepaw in swing at whisker contact binned by the likelihood of the step being lengthened (*Figure 3E*). We compared these to the kinematics we would expect if no modifications were made (dashed gray traces; Materials and methods). A clear pattern emerged: the closer the paw would have landed to the obstacle, the more the step is shortened. However, when the paw would have landed very close to or beyond the obstacle, mice tend to extend the paw to clear it in one large step. This decision-making threshold can be clearly visualized by plotting the landing position probability distributions conditioned on the landing position expected if no modifications are made (*Figure 3F*).

The integrity of this decision-making process depends upon the whiskers. When mice have vision but no whiskers their paw no longer lands cleanly in front of or beyond the obstacle. Rather, the bimodal landing position distribution, which is characteristic of the decision-making process, becomes diffuse (*Figure 3G*). Furthermore, models trained on whiskers-only trials were less accurate when evaluated on light-only trials, suggesting that the decision-making process is meaningfully altered (*Figure 3—figure supplement 1F*, p<0.05, 'W' vs. 'W → L' conditions). Consistent with this interpretation, mice no longer lengthen or shorten their step based on where it would have landed if no modifications were made (*Figure 3H*; the predicted [x axis] and actual landing distance [y axis] are similar, such that the landing position distribution clusters around the unity line). Thus, although mice still make kinematic modifications with vision only, these modifications are less accurate and no longer evince the fast decision process that occurs when whiskers are present. Although we cannot rule out the possibility that vision could be utilized in more naturalistic, head-free circumstances, these results demonstrate that whiskers are sufficient for driving a rapid, body-state dependent sensorimotor transformation.

Finally, mice with neither whiskers nor vision fail to make modifications except when their paws collide with the obstacle (*Figure 3G–H*), and models trained on these trials have reduced accuracy

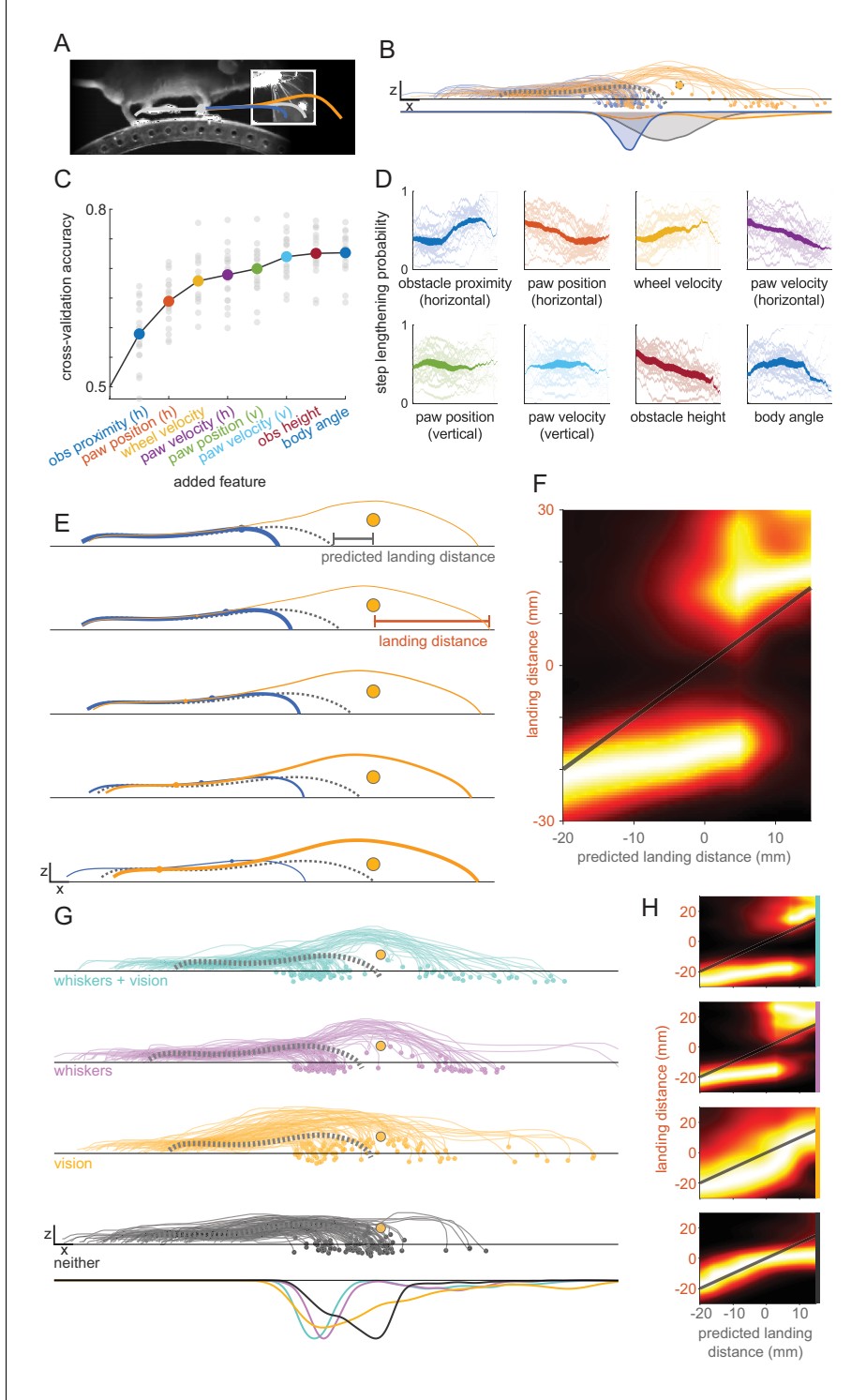

**Figure 3.** A rapid decision underlies obstacle clearance. (**A**). Schematic showing that the paw in swing at whisker contact (gray circle) can be placed in front of the obstacle (blue trace) or extended to clear the obstacle in one step (orange trace). (**B**) Mice shorten or lengthen their step to avoid the obstacle. Each trace shows the kinematics for the paw in swing at whisker contact for a single trial selected randomly across all mice, colored by whether the step was shortened (blue) or lengthened (orange) relative to control steps. The dashed gray trace shows the average control step. Distributions of landing positions (bottom row) reveal that steps are shortened or lengthened relative to control steps. Trials in which the length of the step is unchanged are not included in this plot. (**C**) GLMs accurately predict whether steps are lengthened or shortened. Accuracy is plotted as features are

*Figure 3 continued on next page*

*Figure 3 continued*

added to the model. Models are constructed for each mouse (gray circles) and per-mouse accuracy is the average 15-fold cross-validation accuracy. Features are sequentially added based on their ability to improve the models' average accuracy across mice. h: horizontal; v: vertical; obs: obstacle. (D) The decision varies systematically with both body state and obstacle position. Each plot shows the probability of step lengthening as a function of one predictor used in the model, sorted by the order in which they are included in the model (colors are the same as C). X axes show the 1st to 99th percentile for each predictor. Transparent lines are averages for individual mice, and opaque lines are the averages across mice. Line thickness represents the probability distribution for the predictor. (E) Mice lengthen or shorten their steps based on where the paw would have landed relative to the obstacle. Each row shows the average kinematics for the paw in swing at contact when that paw is placed in front of the obstacle (blue) or clears the obstacle in one step (orange). The rows are binned by the models' confidence that the step will be shortened (top row) vs. lengthened (bottom row). Line thickness is proportional to the likelihood of the step landing in front of vs. over the obstacle. Blue and orange dots show the average position within the trajectory at which whisker contact occurs. (F) Distributions of landing distances (columns) conditioned on where the paw would have landed if no modifications were made ('predicted landing distance'). Predicted landing distance is computed based on running speed and the lift-off position of the paw. (G) Behavioral modifications are more systematic when whiskers are available. Plots are like B, but broken down by sensory condition for the dataset used in *Figure 2* (n = 5) and including trials where no modification is made. (H) Like F, but broken down by sensory condition (rows in H correspond to rows in G). Panels B–F use the same dataset from *Figure 1* (n = 20). See also *Figure 3—figure supplement 1*.

The online version of this article includes the following figure supplement(s) for figure 3:

**Figure supplement 1.** A rapid decision underlies obstacle clearance.

(*Figure 3—figure supplement 1F*, p<0.05), ruling out roles for other sensory modalities. Collectively, these results show that mice lengthen or shorten their steps to avoid collision with the obstacle, choosing between these strategies by integrating information about their body state with sensory information obtained from the whiskers.

## Obstacle clearance intact after barrel cortex lesions

Sensorimotor transformations are commonly thought to involve signals passing from sensory to motor areas of the cerebral cortex (see e.g. *Drew and Marigold, 2015*). Given the importance of whiskers in our task, we lesioned vibrissal primary sensory cortex ('barrel cortex') to examine its role in obstacle clearance behavior (*Figure 4A*, *Figure 4—figure supplement 1A*; Materials and methods). Mice performed the task in complete darkness with whiskers trimmed on one side of the face (*Figure 2—figure supplement 1D–E*). Ipsilateral lesions served as a control because barrel cortex receives information from contralateral whiskers. Because ipsilateral lesions had little impact (*Figure 4—figure supplement 1B*), we analyzed pooled contralateral and bilateral lesions (n = 8).

Barrel cortex lesions had small effects on obstacle clearance the following day that recovered within 4 days (*Video 5*). In animals with contra- or bilateral lesions (n = 8), there were no effects the first day post-lesion on basic locomotion, including running velocity, the height of the body above the wheel, and body angle (*Figure 4B*). Obstacle clearance rates decreased slightly the day immediately following contralateral lesions but recovered to baseline levels within 4 days (*Figure 4C–D*). The early effects appear to be due to decreased paw heights during obstacle clearance, which recovered on the same timescale as success rates (*Figure 4E–F*). Notably, contralateral barrel cortex lesions had no effect on the correlation between leading forepaw and obstacle height (*Figure 4G–H*).

Finally, we trimmed all whiskers to determine whether mice learned to compensate for the absence of barrel cortex by relying on other

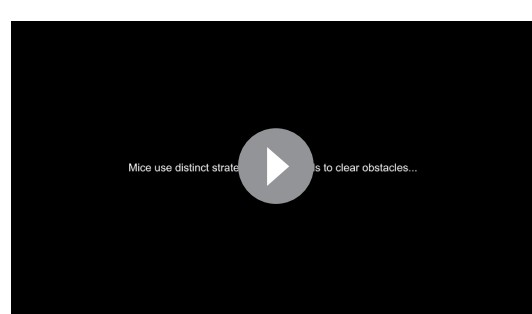

**Video 4.** Demonstration of the two behavioral strategies mice use to clear obstacles. Videos are paused at whisker contact. The dashed gray traces show the kinematic trajectory expected if no modifications are made.
https://elifesciences.org/articles/63596#video4

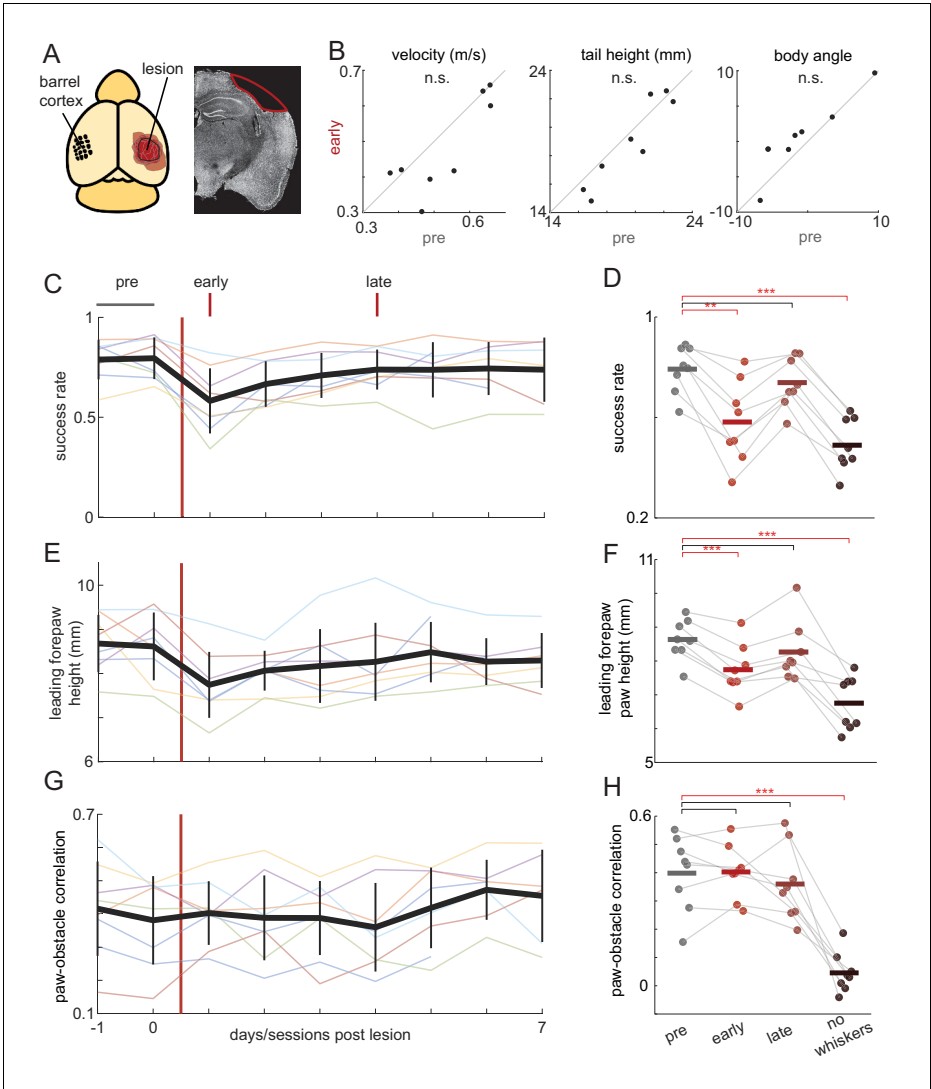

**Figure 4.** Obstacle clearance minimally affected by barrel cortex lesions. (**A**) Schematic showing locations and extent of barrel cortex lesions for all mice (n = 8) and an example coronal section for one mouse. (**B**) Locomotion is unaffected by contralateral barrel cortex lesions. Velocity, tail height, and body angle are similar in the 2 days before the lesion (pre) and the first day post-lesion (early). (**C-H**) Obstacle clearance is minimally affected by barrel cortex lesions. Left column shows how performance changes across days. Thick black lines show the average across mice, vertical black lines show standard deviation, and thin lines show per-mouse averages. Right column compares performance in the two days before the lesion ('pre'), the first day post-lesion ('early'), the fourth day post-lesion ('late'), and after subsequent whisker trimming ('no whiskers'). Success rates and forepaw height had small decreases following the lesion that quickly recovered, whereas whisker trimming significantly affected success rates, leading forepaw heights, and paw-obstacle correlations. See also *Figure 4—figure supplement 1*. The online version of this article includes the following figure supplement(s) for figure 4:

**Figure supplement 1.** Obstacle clearance minimally affected by barrel cortex lesions.

sensory modalities. As with non-lesioned animals (*Figure 2*), whisker trimming significantly decreased success rates (*Figure 4D*), drove decreases in the height of the leading forepaw as it approached the obstacle (*Figure 4F*), and abolished the correlation between the paw and obstacle height (*Figure 4H*). Collectively, these results indicate that barrel cortex is dispensable for this task. Due to the initial decrease in some performance measures, it is impossible to completely rule out involvement of barrel cortex in the pre-lesioned state. However, we suspect that these may be non-specific effects given that similar changes were observed after ipsilateral control lesions (Cohen's

*d* = 0.062 for control lesions vs. 0.133 for contralateral/bilateral lesions; compare *Figure 4C* and *Figure 4—figure supplement 1B*).

## Motor cortex manipulations impair obstacle clearance

Manipulations of motor cortex are known to affect obstacle avoidance in cats (*Beloozerova and Sirota, 1993*; *Drew et al., 1996*) and freely moving rodents (*Asante et al., 2010*). However, the rapid obstacle avoidance described here involves distinct behavioral strategies and may therefore depend on different brain regions. We thus pharmacologically silenced and lesioned motor cortex to determine its necessity in our task.

Silencing forelimb motor cortex impaired obstacle clearance as well as basic aspects of head-fixed locomotion (*Video 6*). After unilateral injections of muscimol in the M1 rostral forelimb area (*Tennant et al., 2011*; *Figure 5A*) mice ran slower, lifted their paws to a lesser extent, and exhibited changes in gait such that the base of the tail was lower and angled contralateral to the side of the injection (*Figure 5B*, left, *Figure 5—figure supplement 1A*; p<0.05 for velocity, body angle, and tail height). Obstacle clearance also suffered considerably. Paws contacted the obstacle more frequently across obstacle heights (*Figure 5C*, left, *Figure 5—figure supplement 1B–C*), both because they were not lifted high enough (*Figure 5E*, *Figure 5—figure supplement 1D–E*) and because they tended to grab the obstacle (*Figure 5—figure supplement 1F*). The correlation between the height of the leading forepaw and that of the obstacle also decreased significantly (*Figure 5D*, left; p<0.01). The effects on paw height were greatest for the contralateral hindlimb (*Figure 5—figure supplement 1D*; p<0.001), and among the forepaws only the contralateral side had significantly decreased success rates (*Figure 5—figure supplement 1B*).

The effects on obstacle clearance persisted even when controlling for changes in baseline locomotion. We identified pairs of manipulated and control trials that were matched in aspects of baseline locomotion for each mouse (the top 20% of trials that were best matched by running velocity, body angle, and tail height at whisker contact [Materials and methods]). In this subpopulation of trials, locomotion in control and manipulated conditions was indistinguishable at whisker contact (*Figure 5B*, right; p=0.64, 0.47, and 0.47, for velocity, body angle, and tail height, respectively). Nonetheless, success rate and paw height deficits remained, consistent with a direct contribution of motor cortex to obstacle clearance (*Figure 5C,E*, *Figure 5—figure supplement 1B–E*; success rate: p<0.05; paw height: p<0.05 for ipsilateral forelimb and p<0.01 for other paws). A slight reduction in the paw-obstacle height correlation was still seen in matched trials, but this trend was not statistically significant (*Figure 5D*, right; p=0.22).

Unilateral lesions affected behavior similarly to muscimol (*Figure 5F–J*, *Figure 5—figure supplement 1*; *Video 6*). There were deficits in success rates, paw heights, and paw height correlations in addition to deficits in baseline locomotion (*Figure 5G–J*, left, *Figure 5—figure supplement 1*). The effects on obstacle clearance remained in subsets of pre- and post-lesion trials matched for characteristics of baseline locomotion (*Figure 5G–I*, right, *Figure 5—figure supplement 1*). These effects on obstacle avoidance are generally consistent with a previous study in freely moving mice (*Asante et al., 2010*). Notably, performance largely recovered over a week (*Figure 5K*). Further

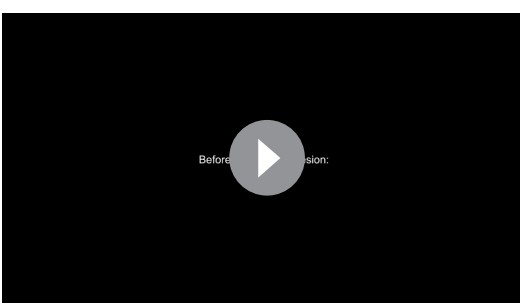

**Video 5.** Obstacle avoidance before and after barrel cortex lesions.
https://elifesciences.org/articles/63596#video5

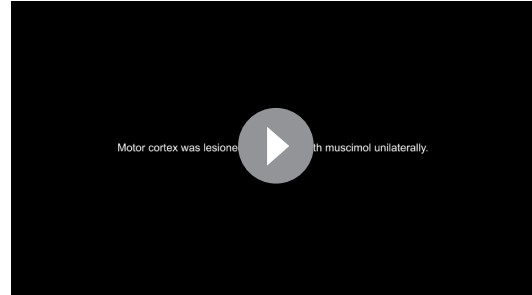

**Video 6.** Obstacle avoidance before and after motor cortex lesions and with injections of saline or muscimol.
https://elifesciences.org/articles/63596#video6

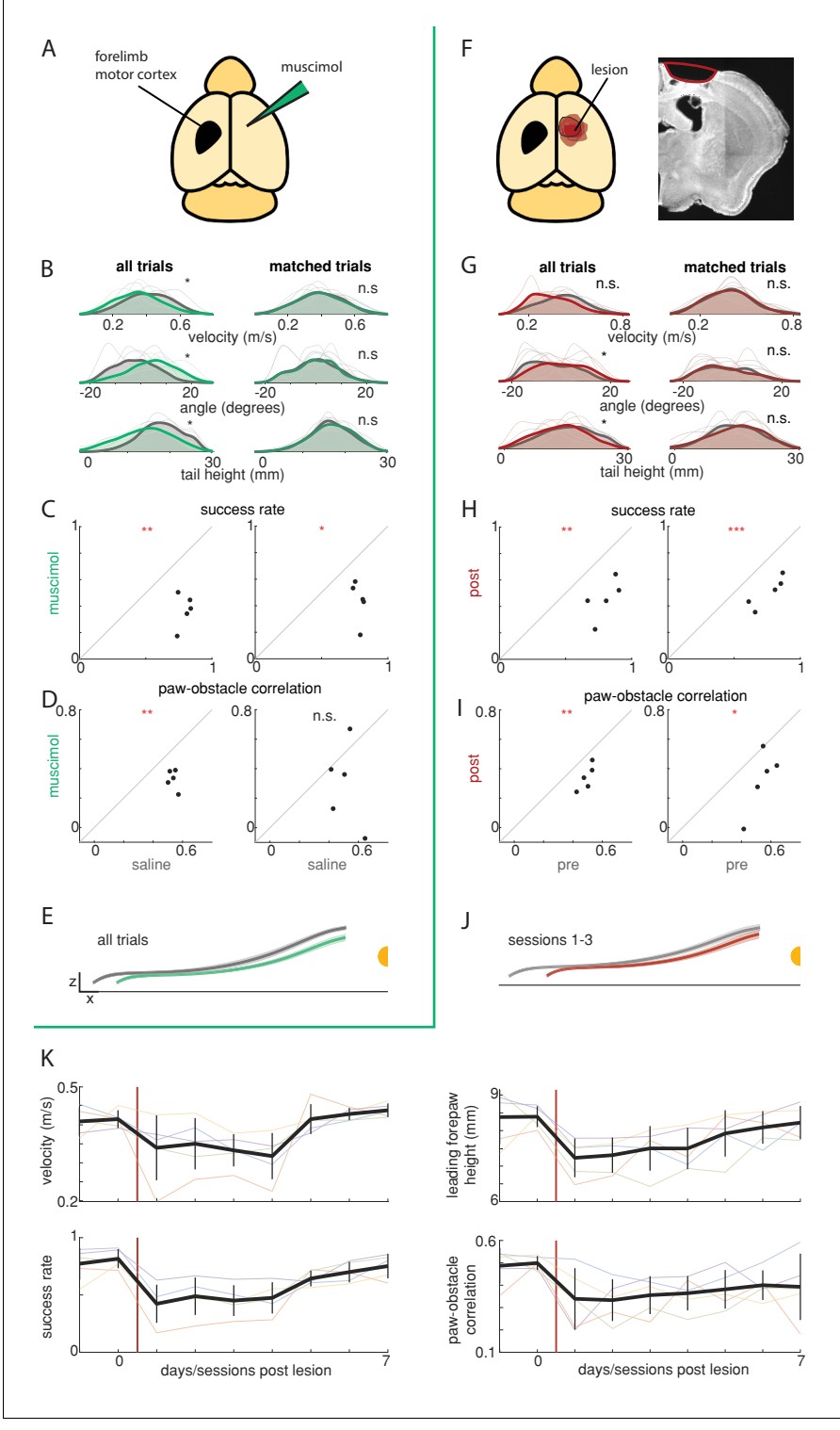

**Figure 5.** Obstacle avoidance affected by motor cortex manipulations. (A–E) Unilateral muscimol injections affect basic characteristics of locomotion as well as obstacle avoidance (n = 5). Left column shows all trials and right column shows 20% of trials selected that are best matched for characteristics of basic locomotion. (B) Distributions for running velocity, body angle, and tail height are matched in the subpopulation of trials. Thin lines show distributions for each mouse in muscimol (green) and saline (gray) conditions. Thick lines show the average distributions across mice, which are very similar following the matching procedure. (C) Mice clear the obstacle at lower rates following muscimol injections (left), even after controlling for changes in locomotion (right). (D) The

*Figure 5 continued on next page*

*Figure 5 continued*

correlation between the height of the leading forepaw and that of the obstacle decreases following muscimol injection (left), although the difference is no longer significant among matched trials (right). (**E**) Kinematics of the leading forepaw as it approaches the obstacle for muscimol (green) and saline (gray) sessions. Paw heights were lower following muscimol injections. Shaded area is standard deviation across mice, and thick lines show the average across individual mouse averages. (**F**) Schematic showing locations of forelimb motor cortex lesions for all mice (n = 5) and an example coronal section for one mouse. (**G-J**) Like B-E, but comparing performance before and after unilateral motor cortex lesions. 'Post' condition shows average performance 1–3 days following the lesion (prior to recovery). Effects on success rates and paw-obstacle correlations persisted in matched trials. (**K**) Performance recovery over time. Thick black lines show the average across mice, vertical black lines show standard deviation, and thin lines show per-mouse averages. See also *Figure 5—figure supplement 1*.

The online version of this article includes the following figure supplement(s) for figure 5:

**Figure supplement 1.** Obstacle avoidance affected by motor cortex manipulations.

---

studies will be required to determine whether the deficits observed immediately after motor cortex manipulations reflect a genuine role for motor cortex in sensory-guided locomotion or acute off-target effects (*Kawai et al., 2015*; *Otchy et al., 2015*).

## Sensorimotor decisions minimally affected by cortical manipulations

We next asked whether manipulations of primary motor and sensory cortices affect the decision to lengthen or shorten strides to clear obstacles.

After motor cortex lesions, the forepaw in swing at whisker contact was still lengthened or shortened to avoid collision with the obstacle (*Figure 6A*), and the body state and obstacle position remained important determinants of the chosen strategy: models trained on pre- and post-lesion sessions were comparably accurate in predicting behavior (*Figure 6—figure supplement 1A*; p=0.06), mice were still more likely to lengthen steps if their paw would have landed closer to the obstacle (*Figure 6B*), and the relationships between obstacle position, body state, and behavioral strategy were similar (*Figure 6C*). We observed the same general pattern of results for muscimol injections into motor cortex, other than a small decrease in model accuracy (6.1% decrease; *Figure 6—figure supplement 1C–G*; p<0.05). The paw in swing at whisker contact tended to land closer to the obstacle in both muscimol and lesion conditions (*Figure 6—figure supplement 1B,D*), an effect that is likely attributable to execution deficits similar to those described above (*Figure 5*). It appears that mice still decide to lengthen or shorten their steps, as evinced by the bimodality of landing position distributions (*Figure 6A,B*), but are less capable of executing the chosen strategy.

Barrel cortex lesions also had minimal impact on the decision-making process. The forepaw in swing at whisker contact was still lengthened or shortened to avoid collision with the obstacle (*Figure 6D*); models trained on pre- and post-lesion sessions were comparably accurate in predicting behavior (*Figure 6—figure supplement 1H*; p=0.67); mice were still more likely to lengthen steps if their paw would have landed closer to the obstacle (*Figure 6E*); and the relationships between obstacle position, body state, and behavioral strategy were similar (*Figure 6—figure supplement 1J*). Lesions caused the forepaw in swing at whisker contact to land closer to the obstacle (*Figure 6—figure supplement 1I*), but this effect mostly recovered after 4 days, the same period over which other small performance deficits recovered (*Figure 4*). Finally, after complete whisker trimming the landing position of the paw was not modified to avoid contact with the obstacle (*Figure 6D*), the paw landed much closer to the obstacle (*Figure 6—figure supplement 1I*), and the accuracy of the models decreased significantly (*Figure 6—figure supplement 1H*; p<0.01). This verifies that mice did not learn to use a different sensory modality to guide decision-making.

## Discussion

We used high-resolution kinematic analysis to characterize sensory-guided locomotion in mice. Using a novel head-fixed assay, we show that mice can rapidly detect and respond to obstacles reliant upon whisker somatosensation. Mice decide how to clear the obstacle – either lengthening or shortening their strides – by integrating information about the location of the obstacle with the position and velocity of their body. Lesions and inactivation of motor cortex impair both baseline locomotion

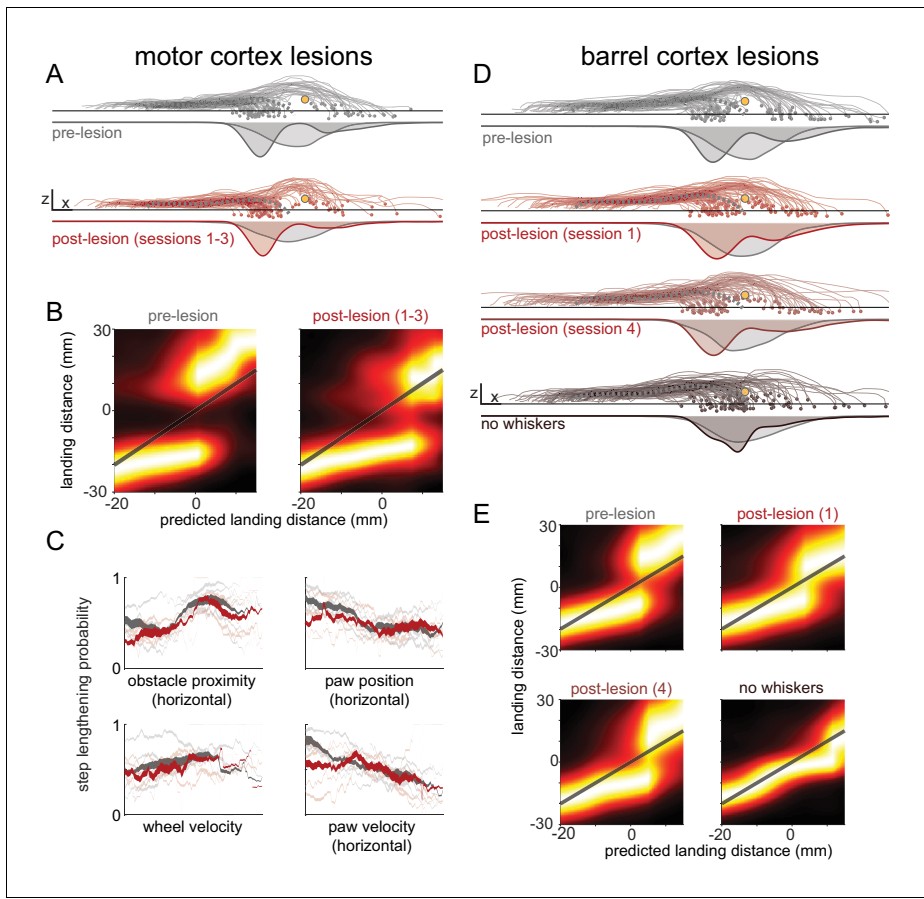

**Figure 6.** Decision-making is minimally affected by cortical manipulations. (A) Mice still shorten or lengthen their step to avoid the obstacle following motor cortex lesions. Each trace shows kinematics for the paw in swing at whisker contact for a trial selected randomly across all mice for either pre-lesion (top row) or post-lesion sessions (bottom row). The thick dashed gray traces show the average control step. Distributions of landing positions are shown beneath the kinematics. (B) Distributions of landing distances (columns within each subplot) conditioned on where the paw would have landed if no modifications were made ('predicted landing distance'). Predicted landing distance is computed based on running speed and the lift-off position of the paw. Mice execute one of two behavioral strategies both before (left) and after (right) lesions. (C) The state of the body and the position of the obstacle are important determinants of whether mice lengthen or shorten their step both before (gray) and after (red) lesions. Each plot shows the probability of step lengthening as a function of one predictor used in the model. X axes show the 1st to 99th percentile for each predictor. Transparent lines are averages for individual mice, and opaque lines are averages across mice. Line thickness represents the probability distribution for the predictor. (D-E) Similar to A-B, but comparing before contralateral barrel cortex lesions, the first day post-lesion, the fourth day post-lesion, and after subsequent whisker trimming. The decision-making process appears to be somewhat affected in the first day post-lesion, but recovers by the fourth day. See also *Figure 6—figure supplement 1*. The online version of this article includes the following figure supplement(s) for figure 6:

**Figure supplement 1.** Decision-making is minimally affected by cortical manipulations.

and obstacle clearance acutely but leave the decision-making process largely intact. Finally, barrel cortex lesions have minimal effects on the behavior, consistent with a subcortical locus for a rapid whisker-based sensorimotor transformation.

## A whisker-mediated sensorimotor transformation independent of barrel cortex

Whisking is important for guiding locomotion in rodents (*Grant et al., 2012*; *Arkley et al., 2014*; *Sofroniew et al., 2014*; *Grant et al., 2018*). Whiskers span the ground where rodents subsequently place their forepaws (*Grant et al., 2018*) and become protracted during high speed locomotion,

suggesting they serve as collision detectors (*Arkley et al., 2014*; *Sofroniew et al., 2014*). Our results support this hypothesis, demonstrating that mice use whiskers to detect obstacles, determine their location, and drive rapid responses to avoid collision even at high speeds.

Although mice had access to visual information several hundred milliseconds prior to whisker contact, the only visually guided anticipatory adjustment we observed was a reduction in running speed on light-on trials. Hence the strategies for obstacle avoidance adopted by mice in the present task contrast with the precise visually guided adjustments of gait, for example of the positioning of the penultimate step, observed in humans (*Higuchi, 2013*) and cats (*Drew et al., 1996*). We cannot rule out that head-fixation and the absence of full-field optic flow in our task disrupted the normal use of vision. Nevertheless, given that mice are largely nocturnal and have relatively poor visual spatial acuity, the rapid whisker-based gait adjustments observed here are likely to be ethologically significant.

What brain regions transmit the whisker information required for obstacle clearance? Barrel cortex neurons are capable of responding to whisker input at short latencies (as fast as 10 ms) (*Crochet and Petersen, 2006*; *Constantinople and Bruno, 2013*; *Yu et al., 2019*), exhibit locomotor-related activity (*Ayaz et al., 2019*), and send projections to subcortical motor areas (*Aronoff et al., 2010*; *Bosman et al., 2011*). Moreover, barrel cortex has been implicated in other forms of whisker-guided locomotion (e.g. wall-tracking) (*Sofroniew et al., 2015*; *Ayaz et al., 2019*). Although recent work has shown that barrel cortex is dispensable for object detection (*Hong et al., 2018*), obstacle clearance additionally requires determining the location of the obstacle by integrating information across multiple whiskers and rapidly adjusting motor output accordingly. Nonetheless, mice were able to perform the task without barrel cortex. Lesions caused minor decreases in success that recovered within 4 days, and the correlation between the height of the leading forepaw and that of the obstacle was unaffected. It cannot be ruled out that the initial performance decrease reflects a role for barrel cortex in the pre-lesioned state. However, control lesions also caused minor decreases in performance, suggesting that the effects of barrel cortex lesions are at least partially attributable to non-specific effects of the manipulation. Further work is needed to resolve this question fully.

The barrel cortex lesion results suggest that subcortical structures are sufficient for transmitting whisker sensory information in this task. Although whisker sensory input is widely distributed in the rodent brain (*Kleinfeld et al., 1999*; *Aronoff et al., 2010*; *Bosman et al., 2011*), two particularly intriguing candidate structures are the cerebellum and the spinal trigeminal nucleus (Sp5). The cerebellum receives extensive whisker somatosensory (*Yatim et al., 1996*; *Rancz et al., 2007*; *Bosman et al., 2011*; *Proville et al., 2014*) and locomotor input (*Arshavsky et al., 1972a*; *Arshavsky et al., 1972b*; *Armstrong, 1986*; *Powell et al., 2015*), projects to subcortical regions involved in locomotor control (*Armstrong, 1986*; *Capelli et al., 2017*; *Fujita et al., 2020*) as well as directly to the spinal cord (*Liang et al., 2011*), has been implicated in normal and skilled locomotion (*Armstrong and Marple-Horvat, 1996*; *Morton et al., 2004*; *Aoki et al., 2013*; *Darmohray et al., 2019*), and has been proposed as a fast route by which sensory-guided behaviors may be coordinated (*Stein and Glickstein, 1992*). Sp5 provides what is likely to be the fastest route between whisker input and motor output; a subpopulation of Sp5 neurons (mainly located within the oralis and interpolaris subdivisions) has been shown to project monosynaptically to forelimb motor neurons (*Esposito et al., 2014*). Although precise patterns of muscle activation may not be computed in Sp5, such a direct pathway appears well suited to rapidly trigger spinal programs for obstacle avoidance, as discussed further below.

## Obstacle avoidance involves a rapid sensorimotor decision

We found that obstacle avoidance involves a rapid sensorimotor decision wherein mice lengthen or shorten their steps depending on the state of the body and a whisker-derived estimate of the obstacle location. Although such sensorimotor decisions are presumably vital for fast locomotion over complex terrain, to our knowledge they have not been thoroughly studied in the laboratory. It is notable that while the logic underlying the decision process is relatively simple, revealing it involved the analysis of a large amount of high-dimensional data (>100,000 trials and >150 million video frames). Hence the present results highlight the power of machine learning for the quantitative analysis of behavior (*Krakauer et al., 2017*; *Datta et al., 2019*).

The cerebral cortex is widely associated with cognitive and perceptual decision-making (*Gold and Shadlen, 2007*; *Cisek and Kalaska, 2010*; *Carandini and Churchland, 2013*) and is

thought to coordinate online responses to perturbations during reaching movements in primates (*Nashed et al., 2014*; *Gallivan et al., 2018*). Nonetheless, cortical manipulations had minimal effects on the decision underlying obstacle clearance. A subcortical locus for the decision is consistent with its rapidity and fits with a rich body of literature supporting the view that even 'low-level' reflexes possess considerable sophistication (*Poppele and Bosco, 2003*; *Rossignol et al., 2006*). Spinal reflexes are adaptively modulated during locomotion by muscle and skin afferents in a manner that depends upon the phase and speed of locomotion (*Grillner, 1975*; *Rossignol et al., 2006*), reminiscent of the phase and speed dependency we observed. Remarkably, even the basic motor program for stepping over obstacles may be present in the spinal cord, as shown by studies of spinalized cats (*Forssberg et al., 1975*; *Forssberg et al., 1977*). Projections from brainstem vestibular nuclei are also known to drive contextually appropriate corrective modifications during locomotion (*Pompeiano and Allum, 1988*; *Murray et al., 2018*). Similarly here, hindbrain pathways may transmit descending whisker signals that are integrated with information about the state of the body within spinal circuits. The decision-making process would thus be defined by feedback rules governing the relationship between whisker sensory input, body state, and subsequent locomotor modifications.

## Materials and methods

### Animals
All experimental protocols were approved by the Columbia University Institutional Animal Care and Use Committee (protocol AC-AABG4566). Adult male wild-type mice (C57BL/6) aged >8 weeks postnatal were used for all experiments. Mice were purchased from Taconic Biosciences (Hudson, NY) and housed in an on-site animal facility on a 12 hr light-dark cycle. Experiments occurred during the light cycle.

### Surgery
Mice received subcutaneous injections of sustained release buprenorphine (0.75 mg/kg) the morning of surgery. Mice were subsequently anesthetized with isoflurane (1.5–2%) and placed in a stereotaxic frame. The skull was exposed and a custom steel headplate (5 × 25 × 1 mm) was attached to the skull with dental cement such that the surface of the headplate was parallel with the horizontal plane connecting bregma and lambda. Mice recovered for 3 days before experiments began.

### Behavioral paradigm and training
Four days after surgery mice began water deprivation and habituation to head fixation on the running wheel. Mice were head fixed on the wheel once daily for about 30 min and were rewarded with drops of water for moving forward on the wheel. On the first day, 0.1–0.3 m of forward movement was rewarded. This amount was increased over ~6 days to 5.4 m (nine wheel rotations), which took ~5–9 days. The position of the head was sometimes fixed further down (ventral) or back (posterior) to encourage running early in training. The head position was gradually adjusted (~1–2 mm per day) until it reached a standard position that was the same for all mice.

Obstacle training then began. Three obstacles were introduced between every reward (*Figure 1—figure supplement 1A*). The movement of the obstacle was matched to the movement of the wheel to simulate moving toward a stationary object. The three obstacles started moving 0.9, 2.7, and 4.5 m after the previous reward, but the positions were jittered ±100 mm (uniformly sampled on this interval) to prevent mice from memorizing the position at which the obstacles arrive. Obstacles were 0.31 m away from the mouse when they started moving; this distance was jittered ±15 mm (uniformly sampled on this interval). The height of the obstacle (the vertical distance between the highest point on the wheel and the highest point of the obstacle) was randomized uniformly across trials between 4 and 10 mm. The behavior occurred in darkness other than a light emitted from the inside of the obstacle that was engaged in a random 50% of trials unless otherwise stated.

To encourage mice to step over (rather than on) the obstacle, it was equipped with a capacitive touch sensor that detected when mice grabbed it. Grabbing the obstacle triggered a white noise auditory stimulus and engagement of a solenoid break that prevented the wheel from moving. Mice were trained daily with obstacles for ~1–2 weeks until performance stabilized, at which point wheel

breaks were rarely triggered (~3% of trials). Experiments then began. Mice unable to run quickly on the wheel (>0.3 m/s after ~2 weeks of training) were excluded from all experiments (~15% of mice). For all training and experiments mice performed one session per day.

## Behavioral apparatus

### KineMouse Wheel

We designed a lightweight (~100 g) running wheel with a transparent floor and mirror mounted inside that allows simultaneous imaging of the side and bottom of the mouse with a single camera (*Warren and Hoffman, 2018*). The wheel consists of a thin polycarbonate floor into which slits were waterjet cut to increase traction and reduce weight. The wheel has lightweight, custom aluminum spokes on one side, and a laser-cut mirror mounted on the other side at 45˚. Wheel motion is captured with an optical rotary encoder.

### Motorized Obstacles

We developed a custom apparatus for controlling obstacles. The obstacle was constructed from a 1/8-inch transparent acrylic rod with white LEDs mounted on either side. The LEDs pointed inside the rod such that light emitted from the surface of the rod when engaged. The side of the obstacle facing away from the mice was coated in copper that served as an electrode for the capacitive touch sensor. The vertical position of the obstacle was set with a linear DC servomotor (Micromo LM0830-015-01) and was controlled with custom Arduino software.

The movement of the obstacle was controlled with a custom belt-driven linear motion system. The obstacle was attached to a platform whose movement was driven by a stepper motor. When the obstacle became engaged, the horizontal movement of the obstacle was matched to the movement of the wheel using custom Arduino software. After the obstacle passed beyond the wheel, it was rotated ~90˚ (using an additional servo motor) and returned to the home position, where it remained before becoming engaged again. The starting distance of the obstacle was approximately 31 cm from the mouse's nose.

### Imaging and data collection

Videos of running and whisking were collected at 250 frames per second using Point Grey Grasshopper (GS3-U3-23S6M-C) and Chameleon (CM3-U3-13Y3M) cameras, respectively. All videos were collected in the dark using infrared illumination. Both cameras were positioned at a large distance (~1.1 m) from the wheel to minimize perspective distortion. Data from all sensors and actuators were recorded with a CED Micro 1401 data acquisition unit. Frame acquisition in both cameras was triggered by TTLs that were also recorded in the 1401, allowing frames to be temporally registered with other data. Frames and metadata from both cameras were acquired using Bonsai acquisition software (*Lopes et al., 2015*).

## Histology

After the final session, mice were anesthetized with ketamine/xylazine and perfused with 4% paraformaldehyde. Brains were sectioned at 100 μm using a cryostat and stained with DAPI. Cortical lesion sizes and locations were determined by tracing lesion boundaries in ImageJ (*Schindelin et al., 2012*) and plotting them on a schematic of the mouse brain using custom MATLAB software.

## Muscimol inactivation

After training mice to perform the task, a ~0.5 mm diameter craniotomy was performed unilaterally over the left or right forelimb motor cortex (1.5 mm lateral, 0.25 mm anterior of bregma) while mice were anesthetized with isoflurane gas. Craniotomies were covered with Kwik-Sil (World Precision Instruments). After 1–2 days of additional training, mice were placed on the wheel while 74 nL total volume of either muscimol (5 μg/μL in saline) or saline was injected at depths of 400 and 700 microns beneath the surface of the brain using a Nanoject II (Drummond). Mice were taken off the wheel for 20 min before the behavioral session commenced. Each mouse received two alternating sessions each of muscimol and saline, with the order counterbalanced across mice (with one session per day). The two sessions for each condition were pooled for all analyses.

## Cortical lesions and barrel field mapping

### Barrel cortex

For barrel cortex lesion experiments, mice performed the task in complete darkness with whiskers remaining on only one side of the face (*Figure 2—figure supplement 1D–E*). This allowed comparison between lesions contralateral and ipsilateral to the remaining whiskers. Since barrel cortex receives information from contralateral whiskers, the ipsilateral lesions served as a control for non-specific effects. Mice received either ipsi- followed by contralateral lesions (n = 3), contra- followed by ipsilateral lesions (n = 1), or contralateral lesions only (n = 4). For mice that received lesions on both sides of the brain, performance was assessed for at least 6 days after the first lesion before the second lesion was performed. Performance was assessed for at least 6 days after the final lesion before trimming all remaining whiskers. Ipsilateral lesions alone had a small impact on success rates that recovered by the second day post-lesion (*Figure 4—figure supplement 1B*); we therefore focused our analysis on pooled contralateral and bilateral lesions.

To avoid damage to nearby somatosensory areas, lesions were targeted by mapping the barrel locations. Barrel field mapping was always conducted on the side of the brain contralateral to the remaining whiskers, and ipsilateral lesions were targeted to the same region on the opposite side of the brain. Intrinsic signal optical imaging of barrel cortex was performed as previously described (*Hong et al., 2018*). Briefly, head-fixed mice were lightly anesthetized with isoflurane while responses to whisker deflection were imaged with a 590 nm long-pass filtered illumination through a thinned skull over barrel cortex. Whiskers were individually deflected with a piezo stimulator at 5 Hz and the corresponding active region was marked using the surface vasculature as a reference. In some cases, the barrels were instead mapped electrophysiologically in isoflurane-anesthetized mice. A ~ 2 mm diameter craniotomy was made around 1.5 mm posterior and 3.2 mm lateral to Bregma. Glass pipettes (3–4 MOhm) were filled with artificial cerebrospinal fluid (ACSF) and inserted into the craniotomy at 350–550 µm below the pial surface. Individual whiskers were manually deflected using a glass Pasteur pipette while amplified and band-pass filtered (0.3–10 kHz) signals were played on an audio monitor to determine the responsive barrel in cortex.

Lesions were performed in trained animals under isoflurane. To avoid damage to other somatosensory areas outside of the barrel fields, the medial barrel field was carefully mapped (delta and E-row) to delineate the barrel field boundaries. A 2–3 mm craniotomy was made according to the mapped barrel fields, and cortical tissue was aspirated using a blunt-tipped needle connected to a vacuum.

### Motor cortex

Motor cortex lesions were stereotactically targeted to the forelimb motor cortex. After the muscimol experiments described above, the same mice were trained for at least 2 days without manipulation. A ~2 mm diameter unilateral lesion was then performed over the forelimb motor cortex (centered at the location of the previous muscimol injection: 1.5 mm lateral, 0.25 mm anterior of bregma). Cortical tissue was aspirated using a blunt-tipped needle connected to a vacuum. Mice were given 200–400 µl of water during recovery before testing the following day for both motor and barrel cortex lesions.

## Behavioral tracking

We used the Kinemouse Wheel (*Warren and Hoffman, 2018*) to capture two orthogonal views of the mouse simultaneously at 250 Hz. We trained a single DeepLabCut network (*Mathis and Warren, 2018*; *Mathis et al., 2018*) to track the positions of body parts and the obstacle in both views, and then stitched the tracking together to reconstruct the body pose in three dimensions. In both the top and bottom view, we tracked all four paws, the base of the tail, the middle of the tail, the nose, and the obstacle.

We initially trained the model on ~200 frames that were labeled using a custom MATLAB GUI. We then manually identified frames with erroneous tracking, corrected these frames, included them in an expanded training set, and retrained the model. This process was repeated until the model was highly accurate and the training set consisted of ~1000 images. The final model's average error was 1.02 pixels (0.27 mm) on the training set and 2.29 pixels (0.60 mm) on the test set.

To further enhance tracking performance, we (1) removed low confidence tracked locations (those beneath a threshold of 0.99), (2) removed tracking when features violated a velocity constraint (i.e. when a feature jumped a large distance in adjacent frames), (3) applied temporal median filtering with a window size of 3 frames, (4) removed tracking when the x position of a feature in the top view was not close to that of the same feature in the bottom view (the x values of the same feature should approximately match because this dimension is shared in the two views), and (5) interpolated the small number of missing values.

To allow direct comparison with freely moving mice, we analyzed kinematics in 'un-head-fixed' coordinates by subtracting the displacement of the wheel (as determined by a high-resolution rotary encoder [U.S. Digital S5-720-250-IE-S-B]) from the kinematic measurements. After this transformation, the position of the obstacle is constant, whereas the mouse moves forward in space.

## Paw contact analysis

### Overview
To detect different types of paw contacts with the obstacle, we built a custom convolutional neural network (CNN) algorithm using Python and fastai (*Howard, 2018*; *Figure 1—figure supplement 1F–H*). First, frames were center cropped around the obstacle in the top view. A ResNet (*He, 2016*)-based CNN classified each subframe as either: no touch, forepaw dorsal touch, forepaw ventral touch, hindpaw dorsal touch, hindpaw ventral (low) touch, or hindpaw ventral (high) touch.

### Image preprocessing
Images were normalized by statistics fitted on ImageNet (*Deng, 2020*) (channel-specific means: [0.485, 0.456, 0.406], standard deviation: [0.229, 0.224, 0.225]).

### Labeling
Training data were labeled by three people using a custom MATLAB GUI. Approximately 30% of trials in 15 sessions were labeled, only including frames when the obstacle is visible. Labelers classified each frame according to the groups listed above.

### Network training
The paw contact network was trained using a transfer learning approach on ResNeXt50 (*Xie, 2017*) pre-trained on ImageNet. A total of 87,239 training frames were used, split into 80% training and 20% validation sets. Data augmentation was performed during training, including random rotation within 10° and lighting changes up to 5%. Training was completed in stages, beginning with scaled-down images of size 42 × 42, then 84 × 84, and ending with the full 168 × 168 images. Categorical cross-entropy loss with class weights was used. Class weights were computed based on the number of training examples per class to compensate for uneven class sizes. The paw contact classifier performed with an overall accuracy of 94.3%, an F1 score of 95%, precision of 96%, and recall of 94% (*Figure 1—figure supplement 1H*).

### Implementation
For each session, cropped images surrounding the obstacle were extracted and processed with the paw contact network. To increase accuracy, test time augmentation was used, wherein four differently augmented versions of each frame were inferenced. Final analysis results were the average of all augmented frame predictions.

The network distinguishes between fore and hind paws, but not left and right paws. To address this, we used DeepLabCut tracking results to determine which paw (left vs. right) was close to the obstacle at each contact frame. For all subsequent analyses, the hindpaw ventral (high) touch class was not used.

### Success determination
Successful trials were those in which there was <= 20 ms of paw contact with the obstacle. When determining the success of individual paws (e.g. *Figure 5—figure supplement 1B*), successful trials were those in which there was no contact with the obstacle.

## Whisker contact analysis

### Overview

We built a custom CNN-based whisker contact algorithm using Python and Keras (*Chollet, 2015*; *Figure 1—figure supplement 1I–K*). Whisker contacts were determined using a high-speed camera focused on the whiskers (*Figure 1A–C*). A combination of LEAP (*Pereira et al., 2019*) and a custom shallow CNN was used to determine the first moment of whisker contact in each trial (*Figure 1—figure supplement 1I–K*). First, LEAP identified the obstacle in the whisker camera. Images cropped around the obstacle were then evaluated on contiguous sequences of 10 frames in sliding windows. For each sequence, a shallow CNN-based network classified the frame within the sequence at which whisker contact first occurred.

### Model description

The location of the obstacle was determined using LEAP. Images were cropped to 200 × 200 pixels around the obstacle and then down sampled to 100 × 100 pixels. The whisker contact network uses a shallow CNN to generate a 5408-dimensional feature vector for each of the 10 contiguous frames (with the same network processing each frame in parallel). These vectors are concatenated and fed into a two-layer fully connected network, which outputs a probability that each frame is the frame of first whisker contact. This analysis is performed in sliding windows across a trial, and the whisker contact frame is determined via consensus across the sliding windows (see below).

The shallow CNN in the whisker contact network is composed of the following layers: 7 × 7 kernel convolution, 32 filters, 2 × 2 stride, same padding; 3 × 3 kernel convolution, 32 filters, 1 × 1 stride, same padding; 2 × 2 kernel max pooling, 2 × 2 stride, same padding; 3 × 3 kernel convolution, 32 filters, 1 × 1 stride, same padding; and flattening (*Figure 1—figure supplement 1J*). All activations are ReLU. The final fully connected network is composed of two layers: a 64-neuron hidden layer with ReLU activation, and a 10-neuron output layer with softmax activation.

### Labeling

A custom Python GUI was used to label the first frame of whisker contact in ~30% of trials from 13 sessions. Training images were then exported for each trial only for frames where the obstacle is visible. This yielded 44,740 training frames, which were split into 80% training and 20% validation sets.

### Training

The whisker contact network, consisting of both the shallow CNN and the two fully connected layers, was trained end-to-end. Inputs were given as 10 consecutive frames. Ground truth labels were encoded as 11-dimensional one-hot vectors. If the frame of first contact was present in the 10 frames, the relative position (1-10) would be set to 1. If the frame of first contact was not present, the 11th position was set to 1. Although the shallow CNN is used to process all 10 frames in parallel, the weights are shared across all instances and are trained simultaneously.

Training was conducted with a batch size of 32 for 25 epochs using the Adam optimizer (*Kingma and Ba, 2015*) with a learning rate of 0.001. Categorical cross-entropy loss was used, with class weighting based on the number of training examples per class to compensate for uneven class sizes. Data augmentation was also applied, including vertical/horizontal translation (<=10 pixels), zoom (<=10%), and rotation (<=10°).

Contact time predictions on the test set had a mean error of −6.6 ms (−1.6 frames at 250 fps) and a standard deviation of 14.3 ms, with negative errors corresponding to early predictions. Train and test error distributions match well, implying good generalization (*Figure 1—figure supplement 1K*).

### Implementation

Sessions were analyzed trial-by-trial. Only frames where the obstacle was close to the mouse were included. Each frame is first passed through the shallow CNN to extract features. Next, the two-layer fully connected network is slid across all applicable frames with stride of 1. A running total of the number of times each frame is determined to be the first contact frame is kept. The frame that is selected the maximum number of times is chosen as the frame of first contact.

## Whisker trimming experiment

In a subset of mice (n = 4), we assessed performance as whiskers were gradually trimmed (*Figure 2—figure supplement 1D*). There were six conditions, beginning with all whiskers intact and ending with only the delta whisker remaining on one side (depicted in *Figure 2—figure supplement 1D*). Specifically, mice were first tested with (1) all whiskers intact; then (2) all whiskers trimmed on one side of the face; then (3) all but columns A-E, rows 1–3, gamma, and delta trimmed; then (4) columns A-C additionally trimmed; then (5) column D additionally trimmed; and finally (6) only the delta whisker remaining. Performance was assessed for a minimum of 3 days in each condition before proceeding to the next condition.

## Locomotion analysis

### Stance determination

Paws were determined to be in stance when the vertical position of the paw was close to the wheel and the horizontal velocity matched that of the wheel. Specifically, for each paw stance was defined as frames were the horizontal velocity was within 0.2 m/s of the wheel velocity, and the vertical position was within 5 mm of the wheel surface. A 20 ms median filter was then applied to debounce the signal for each paw.

### Control steps

For each paw in each trial, control steps were defined as the two latest steps that occurred prior to whisker contact with the obstacle.

### Time to contact estimation

We estimated the amount of time from whisker contact until a paw would intercept the obstacle if no modifications were made (*Figure 3—figure supplement 1A*). For each trial, we measured the most anterior x position across all paws for each frame. For the 100 frames preceding whisker contact (400 ms), we performed a linear fit for this signal as a function of time, then estimated the time at which the position would intercept that of the obstacle.

## Decision-making analysis

All decision-making analyses focus on the forepaw that is in swing at whisker contact. Because the forepaws are typically out of phase with one another (*Figure 1H*), there is usually a single forepaw in swing at whisker contact. However, on some trials both forepaws were in swing, or both were in stance. These trials were excluded from all decision-making analyses.

### Predicted landing distance

For each trial, we estimated where the forepaw in swing at whisker contact would have landed if no modifications were made. The length of steps are highly correlated with running speed (*Machado et al., 2015*) ($r = 0.84 \pm 0.01$; mean ± S.E.M. for the wheel speed – forepaw length correlations across 53 sessions [n = 20 mice; 2–4 sessions per mouse]), which allowed us to predict where the paw would have landed based on the wheel velocity. For each session, we built a linear model for each paw that predicts step length based on running speed. To train the model, we used the control steps immediately preceding whisker contact. We predicted the forepaw landing position for control steps with $5.5 \pm 0.11$ mm mean absolute error (mean ± S.E.M. for 53 sessions [n = 20 mice]). Finally, we used this model to predict where the forepaw in swing at whisker contact would have landed based on the wheel speed and the lift-off position for this paw.

For sessions in which whiskers were fully trimmed we estimated the moment at which contact would have occurred if whiskers were present; this estimate was used to identify the forepaw in swing at 'whisker contact' for subsequent analyses. Relying on a modest correlation ($r = 0.27$) between whisker contact position and running speed, we built linear models relating trial running speed to whisker contact position for each mouse for sessions in which the whiskers were present. These models were used to approximate the times at which whisker contact would have occurred in sessions with no whiskers.

## Behavior modeling

We built logistic GLMs to predict whether the forepaw in swing at whisker contact was lengthened or shortened relative to the predicted landing distance. We measured eight features at the moment of whisker contact that served as inputs to the model: the vertical and horizontal position of the obstacle, the vertical and horizontal position and velocity of the forepaw in swing at contact, wheel velocity, and body angle. To focus on trials where modifications were made, we discarded those in which the paw landed in front of the obstacle and within ±2.5 mm of the predicted landing position. Steps over the obstacle require modifications in the vertical trajectory (*Figure 1*) and were therefore considered to be modified regardless of landing position. Heatmaps (e.g. *Figure 3F*) and kinematic overlays (e.g. *Figure 3G*) included trials with and without modifications to display the overall distribution of landing positions unless otherwise stated.

For each mouse, we built one model per experimental condition, with trials pooled across sessions within a condition (e.g. one model for all muscimol sessions per mouse and another for all saline sessions per mouse). For each model, we performed 15-fold cross validation. The accuracy of the model was taken to be the average accuracy across the 15 models. We weighted trials to compensate for uneven class sizes. A final model trained on all data was evaluated on shuffled targets to establish baseline performance, which hovered around 50%, as expected. We also compared performance to fully connected artificial neural networks with a single hidden layer (100 units), but these models did not perform better than the GLMs.

## Forward feature selection

To determine which features to include in the models we gradually added them based on their ability to improve the models' accuracy. Starting with single predictors, we trained models and assessed their accuracy after adding each of the remaining predictors one at a time. The predictor that caused the greatest average (across mice) increase in accuracy was then included in the model. This process was repeated until all predictors were exhausted. All eight predictors were included in all subsequent models.

## Reaction time analysis

To determine the speed with which mice responded to whisker input we compared the kinematic trajectories of the forepaw in swing at whisker contact to the trajectories that would be expected if no obstacle had been presented (*Figure 3—figure supplement 1C*). We restricted the analysis to trials in which the paw was mid-swing.

For each trial, we used k-nearest neighbors to collect a family of 40 matched control steps that were as close as possible to the pre-contact kinematic trajectory for the trial (control steps were those occurring before contact with the obstacle). Averaging these steps yields an estimate of what the paw would have done if no obstacle were presented. Subtracting the actual trajectory from this estimate gives us a measure of kinematic change (in x, y, and z) as a function of time relative to whisker contact for each trial.

To verify that these differences did not emerge due to failures in our ability to estimate the kinematics that would have occurred if no obstacle were presented, we performed the same analysis on control steps, using the antepenultimate step to predict what would have happened in the penultimate step before whisker contact. There was very little deviation between the predicted and actual trajectories for control steps (black traces in *Figure 3—figure supplement 1C*).

Reaction time was estimated as the moment at which kinematics diverged from the control trajectories. For each trial, we determined the moment at which the trajectory deviates more than 2.5 standard deviations from the population of matched control steps for that trial. This yields a median latency estimate of 24 ms. We emphasize that this is only an estimate of the reaction time; errors are introduced both by the estimation of the moment of whisker contact, as well as the estimation of the kinematics that would have occurred if no obstacle were presented.

## Statistics

All statistical comparisons are paired t-tests, with statistical significance denoted as *p<0.05, **p<0.01, and ***p<0.001. To validate the use of parametric statistics, we used Kolmogorov-Smirnov tests to check distribution normality for paw heights, success rates, wheel velocity, body angle, tail

height, the landing distance of the paw in swing at contact, model accuracies, and the variability of the trailing forepaw landing distance. We performed these tests on the baseline data collected for the 20 mice included in *Figures 1* and *3*. We failed to reject the null hypothesis that the data are normally distributed for all measures.

## Acknowledgements

We thank Tanya Tabachnik for advice on the design of the KineMouse Wheel and the behavioral apparatus; Chris Rodgers and Avner Wallach for advice on the design of the behavioral apparatus; and Caroline Yu for materials and advice used in the construction of the KineMouse Wheel. Support was provided by NIH/NICHD predoctoral fellowship F31DC016816 (RAW); NIH/NINDS R01 NS094659 (RMB); NIH F32 NS084768 (YKH); and Irma T Hirschl Trust (NBS).

## Additional information

### Funding

| Funder | Grant reference number | Author |
| --- | --- | --- |
| National Institutes of Health | F31DC016816 | Richard A Warren |
| National Institutes of Health | R01 NS094659 | Randy Bruno |
| National Institutes of Health | F32 NS084768 | Y Kate Hong |
| Irma T. Hirschl Trust | | Nathaniel B Sawtell |

The funders had no role in study design, data collection and interpretation, or the decision to submit the work for publication.

### Author contributions

Richard A Warren, Conceptualization, Data curation, Software, Formal analysis, Funding acquisition, Investigation, Visualization, Methodology, Writing - original draft, Project administration; Qianyun Zhang, Data curation, Investigation, Project administration, Writing - review and editing; Judah R Hoffman, Investigation; Edward Y Li, Data curation, Software, Investigation, Visualization, Writing - review and editing; Y Kate Hong, Resources, Funding acquisition, Investigation, Visualization, Writing - review and editing; Randy M Bruno, Resources, Funding acquisition, Writing - review and editing; Nathaniel B Sawtell, Conceptualization, Supervision, Funding acquisition, Project administration, Writing - review and editing

### Author ORCIDs

Richard A Warren ![ORCID] https://orcid.org/0000-0002-5335-0691
Qianyun Zhang ![ORCID] https://orcid.org/0000-0002-7172-6648
Judah R Hoffman ![ORCID] http://orcid.org/0000-0003-2010-0597
Nathaniel B Sawtell ![ORCID] https://orcid.org/0000-0002-1859-8026

### Ethics

Animal experimentation: All experimental protocols were approved by the Columbia University Institutional Animal Care and Use Committee (protocol AC-AABG4566).

### Decision letter and Author response

Decision letter https://doi.org/10.7554/eLife.63596.sa1
Author response https://doi.org/10.7554/eLife.63596.sa2

## Additional files

### Supplementary files

• Transparent reporting form

## Data availability

The main dataset used in the paper is available on figshare: https://doi.org/10.6084/m9.figshare.13337435.v2. Raw data are presented in all bar plots.

The following dataset was generated:

| Author(s) | Year | Dataset title | Dataset URL | Database and Identifier |
|---|---|---|---|---|
| Warren RA, Zhang Q | 2020 | Mouse obstacle dataset | https://doi.org/10.6084/m9.figshare.13337435.v2 | Figshare, 10.6084/m9.figshare.13337435.v2 |

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
