## [Decision Letter]

**Acceptance summary:**

Here the authors show that head-fixed, yet freely treadmill-running mice modulate their step length and height to avoid obstacles. With rigorous paw tracking, they discover distinct strategies for rapid on-the-fly adjustments that avoid obstacle collision by either going above or below the obstacle. Obstacle avoidance in this task is whisker-, but not vision-dependent, yet the behavior is independent of the vibrassal primary sensory cortex. The identification of a sensorimotor decision related to a binary-like choice of sending the paw over or in front of the obstacle opens up opportunities to study online behavioral adjustments in head-fixed mice, a highly tractable experimental preparation.

**Decision letter after peer review:**

Thank you for submitting your article "A Rapid Sensorimotor Decision Underlies Skilled Locomotion in Mice Largely Independent of Cerebral Cortex" for consideration by *eLife*. Your article has been reviewed by three peer reviewers, including Jesse H Goldberg as the Reviewing Editor and Reviewer #1, and the evaluation has been overseen by Ronald Calabrese as the Senior Editor.

The reviewers have discussed the reviews with one another and the Reviewing Editor has drafted this decision to help you prepare a revised submission.

Summary:

In their manuscript, “A Rapid Sensorimotor Decision Underlies Skilled Locomotion in Mice Largely Independent of Cerebral Cortex,” Warren et al. present a novel behavioral paradigm to study obstacle avoidance in head-fixed mice, and describe a set of experiments that test the sensory modalities involved and cortical regions important in this behavior. They show that mice modulate their step length and height to avoid obstacles, and that obstacle avoidance is dependent on whisker input, but not visual input. They discover distinct “strategies” (called a decision) for rapid, sensible paw adjustments that avoid obstacle collision by either going above or below the obstacle. Although obstacle avoidance in this task is whisker-dependent, the behavior is independent of the vibrassal primary sensory cortex. The strengths of this paper are the rigorous behavioral analysis and the execution of the task in head-fixed mice, a tractable prep. The identification of a “sensorimotor decision” related to a binary-like choice of sending the paw over or in front of the obstacle opens up opportunities to study rapid, on the fly behavioral adjustments. Even though the biological discoveries in this paper are unsurprising (see below), reviewers agreed that the paper is timely and of high quality on the behavior-side. The data in the paper support the claims and the analyses were mostly clearly presented. We thus generally support this paper for publication provided some concerns are addressed:

Essential revisions:

(1) Framing of the paper

(1a) The authors both implicitly and, in some cases, explicitly compare the strategies employed, and the neural mechanisms involved, in these experiments with those used in cats and humans making gait modifications on the basis of visual information. However, the important differences in the nature of the tasks is not explicitly introduced or discussed. Clearly, the experiments in cats that are referred to in this manuscript are studying the neural mechanisms underlying anticipatory changes in gait that are initiated by visual examination of the environment. Subjects have generally hundreds of milliseconds or seconds between the detection of the object to be avoided and the time that the movement is executed. As such, there is ample time for decisional processes to act. In contrast, in the current experiments, mice have to modify their gait within 63 ms of detection of an obstacle. Even when vision is used as a stimulus (as far as I can tell, see below), it is only available at the last moment. As such the strategies are reactive to the tactile stimulus. This is not to deny that the mice are able to make clearly adaptive modifications of gait in response to these stimuli. However, I think that there is a need to explicitly introduce and discuss these issues. The Discussion in particular should have a section specifically comparing strategies and the possible similarities and differences in the neural mechanisms that may underlie sensorimotor transformations in the two situations.

(1b) The discussion of the lesion results requires more nuance to allow for possible involvement of a lesioned structure when it is intact.

For instance, the authors write:

"The apparent lack of barrel cortex involvement suggests that subcortical structures are responsible for transmitting whisker sensory information in this task"

Strictly speaking, “responsible” should be changed to “sufficient.”

The data show that there is substantial behavioral perturbation following vS1 lesion, but that time yields recovery of behavior. In most mice with intact vS1, however, vS1 is in fact being used for this task, contrary to this statement. Moreover, if Hong / Otchy are anything to go by, transient (optogenetic/pharmacological) vS1 inactivation would presumably perturb the behavior trial-by-trial. If I were studying vS1 in the context of this task, then, I would feel comfortable interpreting its neural activity as, usually, task-relevant. Of course, something clearly happens post lesion (as in other studies, again, of this genre) that allows the animal to compensate for the loss of vS1 (and, presumably, in some animals, this is the initial state pre-lesion; see below), but I think it is very unfortunate that much of the field has interpreted experiments of this nature as demonstrating lack of involvement. The experiments show the possibility of recovery, but if they say anything about involvement in unlesioned mice. The authors do not rule out that vS1 is probably involved in the task when it is intact.

(2) Interpretation and presentation of vision-deprivation experiments

(2a) There is insufficient information in the manuscript to allow the reader to appropriately interpret the experiments involving visual information (or at least, I am unable to find it). In the experiments in which visual information was used in the place of tactile stimuli, when was the visual stimulus available to the mice? Was visual information of the moving rod available to the mice well before the obstacle approached the limbs or was it applied at the last moment, as for the tactile stimulus? Clearly, this information is critical for the interpretation of the results (for example, in the subsections “Obstacle Clearance is Whisker Dependent” and “A Rapid Sensorimotor Decision Underlies Obstacle Clearance”) and the information needs to be clearly presented both in the Results and in the Materials and methods.

(2b) It is unclear how much of the whisker dependence and visual independence relate to mice generally or the specific behavioral conditions. Head restraint will lead to dramatic deficits in gaze control, which will almost certainly reduce the acuity of moving visual targets. The authors should qualify their results on the relative unimportance of vision by clarifying deficits in gaze control that may be associated with head-restraint, i.e. with something like, "We cannot rule out the possibility that under more natural head-free conditions mice would have better gaze control and rely on vision for obstacle avoidance."

(3) Addressing animal-to-animal variability

Figures 4C-H, 5K: These figures illustrate the lesion effect on multiple kinematic/task parameters. The interpretation here reads as the idea that these parameters deviate from (decline in these measures) then return to baseline. True for average, but it seems that effect size is highly variable for individual mice. This should be probed more deeply. Specifically, it appears there is histology for all animals – can the authors relate effect size to lesion extent to see if it is predictive?

---

## [Author Response]

Essential revisions:(1) Framing of the paper(1a) The authors both implicitly and, in some cases, explicitly compare the strategies employed, and the neural mechanisms involved, in these experiments with those used in cats and humans making gait modifications on the basis of visual information. However, the important differences in the nature of the tasks is not explicitly introduced or discussed. Clearly, the experiments in cats that are referred to in this manuscript are studying the neural mechanisms underlying anticipatory changes in gait that are initiated by visual examination of the environment. Subjects have generally hundreds of milliseconds or seconds between the detection of the object to be avoided and the time that the movement is executed. As such, there is ample time for decisional processes to act. In contrast, in the current experiments, mice have to modify their gait within 63 ms of detection of an obstacle. Even when vision is used as a stimulus (as far as I can tell, see below), it is only available at the last moment. As such the strategies are reactive to the tactile stimulus. This is not to deny that the mice are able to make clearly adaptive modifications of gait in response to these stimuli. However, I think that there is a need to explicitly introduce and discuss these issues. The Discussion in particular should have a section specifically comparing strategies and the possible similarities and differences in the neural mechanisms that may underlie sensorimotor transformations in the two situations.

We agree that there are important differences between this and related paradigms that were insufficiently discussed in the original manuscript. Importantly, the obstacle apparatus was designed to give mice time to incorporate visual information into their behavioral strategy. The obstacle becomes visible ~30 cm away from the mouse, giving them 672 ± 394 ms to respond (depending on the speed of locomotion). Mice nonetheless relied on whiskers to perform the task even when vision was present. Thus, the differences in strategy are more likely to reflect biological differences between mice and other species rather than differences in task design. A paragraph has been added to the Discussion specifically addressing the relationship between this and related paradigms.

Furthermore, the non-motorized wheel allows mice to determine the speed at which the task occurs. Therefore, strictly speaking, mice do not have to respond in ~63 ms. Rather, they choose to run at speeds that minimize the utility of preparatory modifications. We were surprised by this behavior initially. We built a long obstacle track to study how distal visual and proximal whisker sensory input are integrated to drive behavior. That mice perform the task at least as well in the dark is a testament to the utility of whiskers as collision detectors capable of driving fast behavioral responses.

Although visual input is made available well in advance of whisker contact, there are important differences between this visual stimulus and the visual information available in more natural settings. As the reviewers pointed out, head fixation certainly limits the usefulness of vision, as mice cannot adjust the angle of their head to focus on the obstacle (although head fixation is likely detrimental to the whisker system as well). Furthermore, in natural settings, full-field visual flow is a rich source of information that may help localize obstacles. Therefore, the sensory dependence experiments tell us more about what can be done with whiskers alone than what can be done with vision under natural circumstances. We raise this important consideration in the Results and Discussion.

(1b) The discussion of the lesion results requires more nuance to allow for possible involvement of a lesioned structure when it is intact.For instance, the authors write:"The apparent lack of barrel cortex involvement suggests that subcortical structures are responsible for transmitting whisker sensory information in this task"Strictly speaking, “responsible” should be changed to “sufficient.”

We agree and have changed “responsible” to “sufficient”.

The data show that there is substantial behavioral perturbation following vS1 lesion, but that time yields recovery of behavior. In most mice with intact vS1, however, vS1 is in fact being used for this task, contrary to this statement. Moreover, if Hong / Otchy are anything to go by, transient (optogenetic/pharmacological) vS1 inactivation would presumably perturb the behavior trial-by-trial. If I were studying vS1 in the context of this task, then, I would feel comfortable interpreting its neural activity as, usually, task-relevant. Of course, something clearly happens post lesion (as in other studies, again, of this genre) that allows the animal to compensate for the loss of vS1 (and, presumably, in some animals, this is the initial state pre-lesion; see below), but I think it is very unfortunate that much of the field has interpreted experiments of this nature as demonstrating lack of involvement. The experiments show the possibility of recovery, but if they say anything about involvement in unlesioned mice. The authors do not rule out that vS1 is probably involved in the task when it is intact.

The data strongly support the conclusion that barrel cortex is not required for the task. We agree, however, that assessing the role of barrel cortex in the pre-lesioned state is much more challenging. Generally speaking, short-term effects that recover over time could reflect some combination of (1) direct involvement of the manipulated brain region in the task followed by compensation by other brain regions, (2) changes in connected brain regions that are themselves involved in the task followed by compensation for these effects (so-called diaschisis), or (3) non-specific effects of surgical manipulation. The reviewers favor interpretation (1), referring to “substantial behavioral perturbation” and stating that for “most mice… vS1 is in fact being used for this task”. We agree that some pre-lesion involvement of vS1 cannot be ruled out, and we state as much in the revised text (subsections “Obstacle Clearance Intact After Barrel Cortex Lesions” and “A whisker-mediated sensorimotor transformation independent of barrel cortex”). However, the data suggest factors (2) and (3) are at a minimum partially responsible for the observed effects.

In the original manuscript we failed to draw attention to control lesions ipsilateral the remaining whiskers. These lesions also cause small effects that recover over time, which may account for ~50% of the decrease seen for the contralateral/bilateral lesions (Cohen’s *d* = 0.062 for control lesions vs. 0.133 for contralateral/bilateral lesions; compare Figure 4C and Figure 4—figure supplement 1B). Importantly, mice were tested the day after the lesion, so non-specific, short-term effects are perhaps unsurprising. The effects of barrel cortex lesions are therefore unlikely to reflect substantial involvement of vS1 even at baseline. We also note that the shaping of paw height to obstacle height, which in our view is an equally important metric of performance on the task as success, was completely unaffected by barrel cortex lesions. The text has been revised to emphasize this (subsection “Obstacle Clearance Intact After Barrel Cortex Lesions”).

The reviewers also raise the interesting possibility that animals may vary in the extent to which barrel cortex is utilized in the task, thus explaining across-mouse variability in lesion effects. Alternatively, the variability may be due to differences in the extent of the lesions or non-specific differences in the ability of mice to recover immediately from surgical manipulations. Although we cannot resolve this question with the current data (see below), we agree with the reviewers that vS1 involvement cannot be ruled out. We raise this point in both the Results and Discussion, and draw attention to the control lesions such that readers have the opportunity to draw their own conclusions.

(2) Interpretation and presentation of vision-deprivation experiments(2a) There is insufficient information in the manuscript to allow the reader to appropriately interpret the experiments involving visual information (or at least, I am unable to find it). In the experiments in which visual information was used in the place of tactile stimuli, when was the visual stimulus available to the mice? Was visual information of the moving rod available to the mice well before the obstacle approached the limbs or was it applied at the last moment, as for the tactile stimulus? Clearly, this information is critical for the interpretation of the results (for example, in the subsections “Obstacle Clearance is Whisker Dependent” and “A Rapid Sensorimotor Decision Underlies Obstacle Clearance”) and the information needs to be clearly presented both in the Results and in the Materials and methods.

We thank the reviewers for pointing out the lack of clarity on this aspect of the design. As discussed above, the apparatus was designed to make the obstacle visible from a distance (~30 cm). This has been clarified in the text.

(2b) It is unclear how much of the whisker dependence and visual independence relate to mice generally or the specific behavioral conditions. Head restraint will lead to dramatic deficits in gaze control, which will almost certainly reduce the acuity of moving visual targets. The authors should qualify their results on the relative unimportance of vision by clarifying deficits in gaze control that may be associated with head-restraint, i.e. with something like, "We cannot rule out the possibility that under more natural head-free conditions mice would have better gaze control and rely on vision for obstacle avoidance."

We agree that this is an important point. As discussed above, vision is almost certainly more useful under natural conditions. However, that mice can perform the task so skillfully with whiskers alone, even under head-fixation, suggests that whiskers are likely important for sensory guided locomotion. We clarify these points in the text.

(3) Addressing animal-to-animal variabilityFigures 4C-H, 5K: These figures illustrate the lesion effect on multiple kinematic/task parameters. The interpretation here reads as the idea that these parameters deviate from (decline in these measures) then return to baseline. True for average, but it seems that effect size is highly variable for individual mice. This should be probed more deeply. Specifically, it appears there is histology for all animals – can the authors relate effect size to lesion extent to see if it is predictive?

We agree that this is an interesting avenue to explore. To address this, we correlated the lesion area with changes in success rate, paw height, and paw-obstacle correlations. Author response image 1 shows the difference in these measurements between the session before and the first session after barrel cortex lesions (which occurred the day after the lesion).

There is a trend towards a correlation – such that larger lesions are associated with larger effects – but these changes do not approach significance. Ultimately, a much larger dataset would be necessary to determine whether the lesion area relates systematically to the behavioral effects. We performed the same analysis for the motor cortex lesions, which were less variable in size. No trends emerged (see Author response image 2).

**Author response image 2. respfig2:**